# Causal Discovery from Time Series with Hybrids of Constraint-Based and Noise-Based Algorithms

**Daria Bystrova**[*]         *daria.bystrova@univ-grenoble-alpes.fr*
*Univ. Grenoble Alpes, CNRS, Inria, Grenoble INP, LECA, LJK, LIG, F38000, Grenoble, France*

**Charles K. Assaad**[*]         *charles.assaad@inserm.fr*
*Sorbonne Université, INSERM, Institut Pierre Louis d'Epidémiologie et de Santé Publique, F75012, Paris, France*
*EasyVista, F38000, Grenoble, France*

**Julyan Arbel**
*Univ. Grenoble Alpes, CNRS, Inria, CNRS, Grenoble INP, LJK, F38000, Grenoble, France*

**Emilie Devijver**
**Eric Gaussier**
*Univ Grenoble Alpes, CNRS, Grenoble INP, LIG, F38000, Grenoble, France*

**Wilfried Thuiller**
*Univ. Grenoble Alpes, Univ. Savoie Mont Blanc, CNRS, LECA, F38000, Grenoble, France*

**Reviewed on OpenReview:** *https://openreview.net/forum?id=PGLbZpVk2n*

## Abstract

Constraint-based methods and noise-based methods are two distinct families of methods proposed for uncovering causal graphs from observational data. However, both operate under strong assumptions that may be challenging to validate or could be violated in real-world scenarios. In response to these challenges, there is a growing interest in hybrid methods that amalgamate principles from both methods, showing robustness to assumption violations. This paper introduces a novel comprehensive framework for hybridizing constraint-based and noise-based methods designed to uncover causal graphs from observational time series. The framework is structured into two classes. The first class employs a noise-based strategy to identify a super graph, containing the true graph, followed by a constraint-based strategy to eliminate unnecessary edges. In the second class, a constraint-based strategy is applied to identify a skeleton, which is then oriented using a noise-based strategy. The paper provides theoretical guarantees for each class under the condition that all assumptions are satisfied, and it outlines some properties when assumptions are violated. To validate the efficacy of the framework, two algorithms from each class are experimentally tested on simulated data, realistic ecological data, and real datasets sourced from diverse applications. Notably, two novel datasets related to Information Technology monitoring are introduced within the set of considered real datasets. The experimental results underscore the robustness and effectiveness of the hybrid approaches across a broad spectrum of datasets.

## 1 Introduction

Recent technological advances allow collecting observational time series on complex dynamical systems in various fields, such as biodiversity monitoring in ecology (Dornelas et al., 2018), epidemiology (Meci et al., 2022, Arlegui et al., 2023, Bales et al., 2023, Moreau et al., 2023), healthcare (Morid et al., 2023) and Information Technology (IT) monitoring systems (Tamburri et al., 2020, Assaad et al., 2023). One of the

---

[*]These authors contributed equally to this work.

key objectives in studying such dynamical systems is to understand the causal relationships between the system's components. To find these causal relations, experts can employ causal discovery methods for time series, which aim to build a causal graph from observational data. These methods can be categorized into several families, including Granger-based (Granger, 1969), constraint-based (Spirtes et al., 2000, Runge, 2020, Assaad et al., 2022c), score-based (Chickering, 2002b), continuous optimization-based (Zheng et al., 2018) and noise-based families (Hyvärinen et al., 2008, Peters et al., 2013) —for more details see Assaad et al. (2022a), Hasan et al. (2023), Gong et al. (2023). Each family has its own set of assumptions, which may or may not be suitable for a specific dataset. Therefore, no single method stands out in all situations (Assaad et al., 2022a).

Hybrid frameworks combine several methods from different families to enhance graph inference (Hasan et al., 2023). For non-temporal data, several authors propose to combine ideas from constraint-based and score-based methods to improve scalability (Tsamardinos et al., 2006) or robustness to small sample size (Ogarrio et al., 2016). For temporal data, SVAR-GFCI, proposed by Malinsky & Spirtes (2018), is a time-series generalization of the hybrid method GFCI which is based on the score-based and constraint-based algorithms.

Another type of hybrid framework, which is our main focus in this work, is based on the combination of constraint-based and noise-based families. The advantage of constraint-based methods is that they are non-parametric (i.e., no assumption is made on the form of the underlying causal relationships), while the limitation is that they require strong non-testable assumptions and can only recover the causal graph up to its Markov equivalence class, i.e., orientation of some edges could be unknown in the inferred graph as several graphs represent the same conditional dependence structure. On the other hand, noise-based methods are capable of recovering true graphs. So, by combining methods from both families, although we require assumptions of both families of methods, some assumptions can be weakened and we can recover the true causal graph. There exist several methods of this type for non-temporal data, such as PClingam (Hoyer et al., 2008) or FRITL (Chen et al., 2021). Assaad et al. (2021) introduced NBCB$^{\text{acyclic}}$ [1] for temporal data assuming that there are no cyclic causal relations between different time series.

This paper presents a hybrid framework for temporal data using noise-based and constraint-based algorithms. In this framework, we consider two different classes of methods, which we denote NBCB and CBNB. Both classes can infer different types of causal graphs that differentiate between instantaneous relations and lagged relations. To construct these types of causal graphs, NBCB and CBNB orient edges using a noise-based strategy and prune edges using a constraint-based strategy. The main difference between NBCB and CBNB is that the former starts by orienting the graph and then proceeds to pruning, while the latter starts by pruning and then proceeds to orientation. Most, importantly, NBCB and CBNB combine the parts from corresponding methods in an efficient way, such that information in the first part improves the efficiency of the second part. At the core of CBNB lies the notion of an undirected cycle group, which we introduce to optimize the search for orientation. The advantage of the proposed hybrid framework is that, in practice, it has a trade-off performance between the constraint-based and noise-based algorithms. In comparison to the constrained-based family, the proposed methods do not require the so-called faithfulness assumption (but require a weaker assumption named adjacency faithfulness) and can recover the true causal graph instead of restricting only to the markov equivalence class. In comparison to noise-based methods, the proposed methods provide a better pruning of edges when the sample size is small (Malinsky & Danks, 2018). In this paper, we also show the distinct responses of NBCB and CBNB when assumptions are violated. This demonstration highlights the expected results across various scenarios and aids in determining the most suitable methodological approach for a given problem.

In summary, our main contributions are the following:

- We propose a hybrid framework for the causal discovery of time series that combines parts of noise-based and constraint-based algorithms. Within this framework, we derive two classes of algorithms, NBCB and CBNB, which we optimize to infer the causal graph from time series.

---

[1]In Assaad et al. (2021), this method was called NBCB but in this work, we denote it as NBCB$^{\text{acyclic}}$ to explicitly point out that it assumes that the summary causal graph is acyclic. For more details about summary causal graphs see Section 2.

$$X_t := a_x X_{t-1} + a_{yx} Y_t + a_{zx} Z_t + \xi_t^x$$

$$Z_t := a_z Z_{t-1} + a_{xz} X_{t-1} + a_{yz} Y_t + a_{wz} W_{t-1} + \xi_t^z$$

$$Y_t := a_y Y_{t-1} + a_{xy} X_{t-1} + a_{wy} W_{t-2} + \xi_t^y \qquad (1)$$

$$W_t := a_w W_{t-1} + a_{yw} Y_t + a_{zw} Z_t + \xi_t^w$$

$$U_t := a_u U_{t-1} + a_{wu} W_t + \xi_t^u.$$

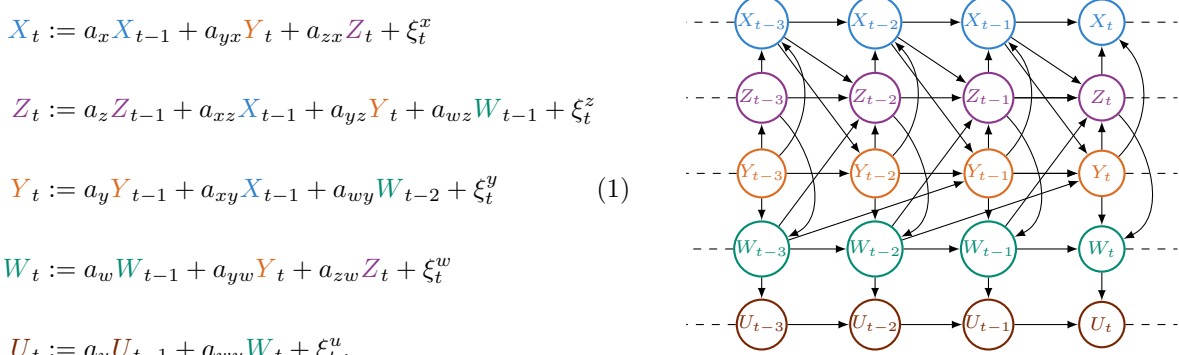

Figure 1: Running example. Left: Dynamic structural causal model (dynamic SCM). Right: Associated full-time causal graph $\mathcal{G}^{\mathrm{f}}$ (FTCG).

- We study theoretically to which extent each class of algorithms is robust against assumption violation.

- We provide extensive simulation studies and real data applications to illustrate the applicability of our approach and their enhanced capabilities against assumption violation compared to original methods.

- We introduce two novel datasets about IT monitoring within the set of considered real datasets.

The remainder of the paper is organized as follows: Section 2 describes the different types of causal graphs that can be used to represent causal relations between time series and the different assumptions related to those graphs. Section 3 discusses related work and particularly details the steps that compose noise-based and constraint-based algorithms. Section 4 introduces our main contribution, the hybrid framework, which consists of two classes NBCB and CBNB, each of which is detailed in dedicated subsections. In Section 5, NBCB and CBNB are compared to different causal discovery algorithms on simulated, realistic, and real datasets. Finally, Sections 6 and 7 discuss and conclude the paper.

## 2 Background

In this section, we first introduce some terminology, tools, and assumptions which are standard for the major part. We use upper case letters to denote observed random variables, lower case letters to represent deterministic constants, blackboard bold for sets, and Greek letter $\xi$ to denote noise. A directed graph is denoted as $\mathcal{G}$ and parents and descendants of $X$ in $\mathcal{G}$ are respectively denoted as $\mathrm{Pa}_{\mathcal{G}}(X)$ and $\mathrm{Desc}_{\mathcal{G}}(X)$. We denote $\langle X, \ldots, Y \rangle$ a path in a graph which starts at node $X$ and ends at node $Y$ and we denote $\langle X - Z, \ldots, W - Y \rangle$ a walk in a graph which starts at node $X$ and ends at node $Y$ (nodes are used for paths and edges are used for walks). The skeleton of a directed graph, is a graph that consists only of undirected edges that represent the same adjacencies as in the directed graph (Hasan et al., 2023).

Causal relations in a dynamical system can be represented by a dynamic structural causal model, an extension of structural causal model (SCM, Pearl, 2000) to time series. In such dynamic SCM, each point in a time series is defined by the function of its parents and some unobserved noise. Without loss of generality, we will use the linear dynamic SCM represented in Equation (1) as a running example.

Such a dynamic SCM can be represented graphically using a full-time causal graph (FTCG, Assaad et al. (2022a)), as represented in Figure 1 and denoted as $\mathcal{G}^{\mathrm{f}} = (\mathbb{E}^{\mathrm{f}}, \mathbb{V}^{\mathrm{f}})$. The FTCG represents an infinite graph of the dynamical system through infinite nodes $\mathbb{V}^{\mathrm{f}}$, representing observational random variables, and infinite edges $\mathbb{E}^{\mathrm{f}}$, representing causal relationships. In this paper, we only consider FTCGs that are directed acyclic graphs (DAGs), i.e., all edges are directed and there exists no directed path in $\mathcal{G}^{\mathrm{f}}$ that starts and ends at the same node.

### 2.1 Assumptions

We assume that there are no hidden common causes, an assumption known as causal sufficiency.

**Assumption 1** (Causal sufficiency, Spirtes et al., 2000). *Let $\mathcal{G}^{\mathrm{f}} = (\mathbb{E}^{\mathrm{f}}, \mathbb{V}^{\mathrm{f}})$ be an FTCG. There exist no hidden common causes of any two observed nodes in the $\mathbb{V}^{\mathrm{f}}$, i.e., the noise terms in the underlying dynamic SCM are jointly independent.*

One of the most common assumptions is the causal Markov condition, which is assumed by most of the methods, and connects the causal graphs that correspond to the given SCM with the compatible probability distribution.

**Assumption 2** (Causal Markov condition, Spirtes et al., 2000). *Let $\mathcal{G}^{\mathrm{f}} = (\mathbb{E}^{\mathrm{f}}, \mathbb{V}^{\mathrm{f}})$ be an FTCG and $P$ be a probability distribution over the nodes in $\mathbb{V}^{\mathrm{f}}$ generated by the causal structure represented by $\mathcal{G}^{\mathrm{f}}$. Every $X \in \mathbb{V}^{\mathrm{f}}$ is independent of $\mathbb{V}^{\mathrm{f}} \backslash \{\mathrm{Desc}_{\mathcal{G}^{\mathrm{f}}}(X) \cup \mathrm{Pa}_{\mathcal{G}^{\mathrm{f}}}(X)\}$ conditional on $\mathrm{Pa}_{\mathcal{G}^{\mathrm{f}}}(X)$.*

In general, inferring causal graphs from data is possible under additional assumptions on the data-generating process. For the constraint-based family of methods, the necessary assumption for the correspondence between the graph and the distribution is the faithfulness assumption (Spirtes et al., 2000), which states that all the conditional independence relations that are true in the probability distribution are entailed by the causal Markov condition applied to $\mathcal{G}^{\mathrm{f}}$. However, we consider in this work the following weaker version of the faithfulness assumption, called adjacency faithfulness.

**Assumption 3** (Adjacency Faithfulness, Ramsey et al., 2006). *Let $\mathcal{G}^{\mathrm{f}} = (\mathbb{E}^{\mathrm{f}}, \mathbb{V}^{\mathrm{f}})$ be an FTCG. If two nodes $X$ and $Y$ in $\mathbb{V}^{\mathrm{f}}$ are adjacent in $\mathcal{G}^{\mathrm{f}}$, then they are dependent conditionally on any subset of $\mathbb{V}^{\mathrm{f}} \backslash \{X, Y\}$.*

We provide an example of the violation of the adjacency faithfulness and faithfulness assumptions in Figure 2. Let us consider variables $W_t$ and $U_t$ and their past $(W_{t-1}, U_{t-1}, \ldots)$ in the causal graph in Figure 2a. Let us assume that the corresponding linear SCM is composed of two equations:

$$W_t = a_w W_{t-1} + \xi_t^w$$
$$U_t = a_u U_{t-1} + a_{wu} W_t + \xi_t^u,$$

which leads to

$$U_t = a_u^2 U_{t-2} + (a_u a_{wu} + a_w w_{wu}) W_{t-1} + a_{wu} \xi_t^w + a_u \xi_{t-1}^u + \xi_t^u.$$

If the coefficients are such that $a_u = -a_w$, then the coefficient before $W_{t-1}$ is 0 and thus $W_{t-1} \perp\!\!\!\perp U_t \mid U_{t-2}$ (the noise terms are jointly independent and independent of $U_{t-2}$ and $W_{t-1}$). If there exists such combination of coefficients, we say that the path $\langle W_{t-1}, W_t, U_t \rangle$ is canceled by the path $\langle W_{t-1}, U_{t-1}, U_t \rangle$ (or vice versa). The independence $W_{t-1} \perp\!\!\!\perp U_t \mid U_{t-2}$ is not entailed by the the causal Markov condition, so the faithfulness assumption is violated, while adjacency faithfulness is not violated as $W_{t-1}$ and $U_t$ are not adjacent. Let us now consider the graph with an extra edge as designed in Figure 2b. If the path $\langle W_{t-1}, U_t \rangle$ is canceled out by the two paths $\langle W_{t-1}, W_t, U_t \rangle$ and $\langle W_{t-1}, U_{t-1}, U_t \rangle$ then $W_{t-1} \perp\!\!\!\perp U_t \mid \{U_{t-2}, W_{t-2}\}$ and the adjacency faithfulness assumption is violated. Notice that $W_{t-1} \perp\!\!\!\perp U_t \mid \{U_{t-2}, W_{t-2}\}$ is not entailed by the the causal Markov condition, which means that the faithfulness assumption is also violated.

Note that adjacency faithfulness is proven to be equivalent to the minimality condition (Peters et al., 2017) introduced in Spirtes et al. (2000), which states that every proper subgraph of $\mathcal{G}^{\mathrm{f}}$ does not satisfy causal Markov condition.

In addition, the following assumption is required by the noise-based family to guarantee identifiability.

**Assumption 4** (Identifiable Functional Model, Peters et al., 2011). *The SCM belongs to an* identifiable *functional model* class, as defined in *Peters et al. (2011).*

For example, the linear dynamic SCM in Equation (1) satisfies Assumption 4 if the noise terms are non-Gaussian.

### 2.2 Causal graphs for time series

Using time series data has a particular advantage for discovering causal relationships between temporal variables since we can employ temporal priority, which states that the causal relationship between two variables

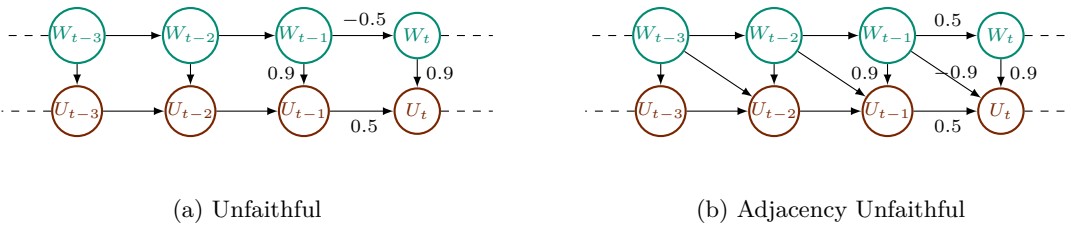

(a) Unfaithful

(b) Adjacency Unfaithful

Figure 2: Illustration of the violation of the faithfulness and the adjacency faithfulness assumption. (a) violates the faithfulness assumption but satisfies the adjacency faithfulness assumption and (b) violates the adjacency faithfulness which implies that it violates the faithfulness assumption.

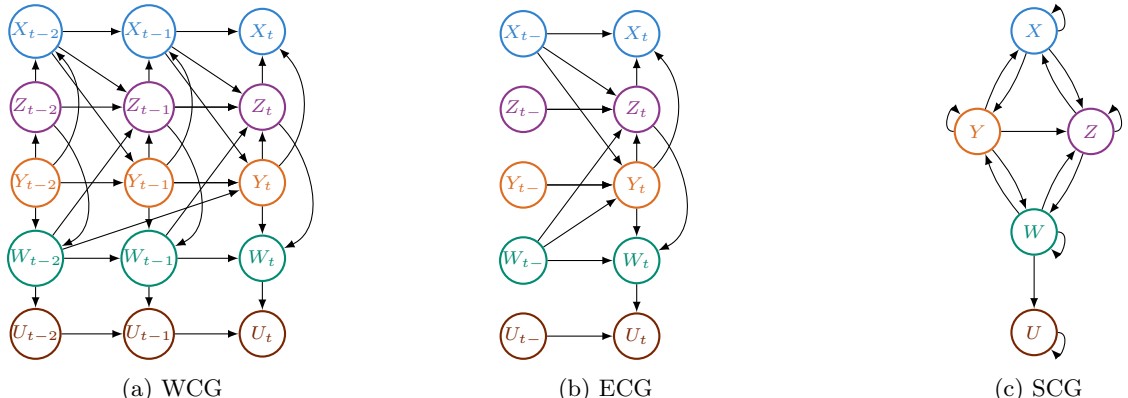

(a) WCG

(b) ECG

(c) SCG

Figure 3: Different causal graphs to represent the dynamic SCM in Equation (1): (a) window causal graph (WCG) with a maximal temporal lag equal to 2, (b) extended summary causal graph (ECG), and (c) summary causal graph (SCG).

is oriented such that the effect cannot happen before its cause. However, despite this advantage, working with full-time causal graphs (which was introduced before) is impractical due to their infinite dimension, which has led to the adoption of simpler causal graphs, assuming that causal relations between time series hold throughout time. This is formalized in the following assumption, in which $\mathbb{V}^{\mathrm{f}} = (\mathbb{V}_{-\infty}, \ldots, \mathbb{V}_t, \ldots, \mathbb{V}_{\infty})$ where $\mathbb{V}$ denotes a vector of $d$ time series.

**Assumption 5** (Consistency Throughout Time). *Let $\mathcal{G}^{\mathrm{f}} = (\mathbb{E}^{\mathrm{f}}, \mathbb{V}^{\mathrm{f}})$ be a full-time causal graph. There exists $\gamma$ in $\mathbb{N} \setminus \{0\}$ such that the causal structure of the graph consisting of the nodes $\{\mathbb{V}_{t-\gamma}, \ldots, \mathbb{V}_t\}$ is the same for every $t$.*

We call the minimum value of $\gamma$ for which consistency throughout time holds the *maximal temporal lag* of the graph. For example, the maximal temporal lag of the graph of Figure 1 is $\gamma = 2$.

Under Assumption 5, the full-time causal graph is equivalent to the so-called window causal graph (Hyvärinen et al., 2008, Runge, 2020, Figure 3a).

**Definition 1** (Window Causal Graph, WCG). *Let $\mathcal{G}^{\mathrm{f}} = (\mathbb{E}^{\mathrm{f}}, \mathbb{V}^{\mathrm{f}})$ be a full-time causal graph satisfying Assumption 5 with $\gamma$ the maximal temporal lag in $\mathcal{G}^{\mathrm{f}}$. A window causal graph (WCG) $\mathcal{G}^{\mathrm{w}} = (\mathbb{E}^{\mathrm{w}}, \mathbb{V}^{\mathrm{w}})$ is the subgraph of $\mathcal{G}^{\mathrm{f}}$ consisting of the nodes $\mathbb{V}^{\mathrm{w}} = (\mathbb{V}_{t-\gamma}, \ldots, \mathbb{V}_t)$ and $\mathbb{E}^{\mathrm{w}}$ contains all related edges.*

Unfortunately, causal discovery methods still suffer in practice of the strong assumptions they rely on which are not always satisfied. Thus, in many applications, experts have to validate those graphs before using them. However, validating WCGs is challenging: if experts can usually identify causes and related effects, they do not know, in general, the exact temporal lags between them. Moreover, in some applications, experts are not interested in understanding exact causal relations with the temporal lags between them but rather opt for an abstract representation of the causal relationships between variables. An extended summary causal graph

(Assaad et al., 2022c, Figure 3b) is one type of abstraction that can differentiate between instantaneous relations and lagged relations without giving precise information about the lag.

**Definition 2** (Extended Summary Causal Graph, ECG). *Let $\mathcal{G}^{\mathrm{w}} = (\mathbb{E}^{\mathrm{w}}, \mathbb{V}^{\mathrm{w}})$ be a WCG with maximal temporal lag $\gamma$ and nodes $(\mathbb{V}_{t-\gamma}, \ldots, \mathbb{V}_t)$. The underlying extended summary causal graph (ECG) $\mathcal{G}^{\mathrm{e}} = (\mathbb{E}^{\mathrm{e}}, \mathbb{V}^{\mathrm{e}})$ consists of the nodes $\mathbb{V}^{\mathrm{e}} = (\mathbb{V}_{t-}, \mathbb{V}_t)$ and the set of directed edges $\mathbb{E}^{\mathrm{e}}$ is defined as follows: for all $X_t, Y_t \in \mathbb{V}_t$, $X_t \neq Y_t$, there exists a directed edge from $X_t$ to $Y_t$ (denoted as $X_t \rightarrow Y_t$) if and only if the same directed edge exists in $\mathcal{G}^{\mathrm{w}}$; for all $X_{t-} \in \mathbb{V}_{t-}, Y_t \in \mathbb{V}_t$, there exists a directed edge from $X_{t-}$ to $Y_t$ (denoted as $X_{t-} \rightarrow Y_t$) if and only if there exists at least one temporal lag $\ell > 0$ such that there exists a directed edge between $X_{t-\ell}$ and $Y_t$ in $\mathcal{G}^{\mathrm{w}}$.*

The ECG presents several interesting characteristics: similarly to the WCG, it inherits the acyclicity of the FTCG; it is possible to infer it from data without passing a WCG without any additional assumption (it only needs the assumptions that are needed to infer a WCG); the ECG and the WCG are equivalent when the maximal temporal lag $\gamma$ is 1.

Another more abstract graphical representation is the summary causal graph (Arnold et al., 2007, Peters et al., 2013, Assaad et al., 2022a, Figure 3c), which provides an overview of the causal relationships between time series without any information about the temporal lag (it does not differentiate between instantaneous relations and lagged relations).

**Definition 3** (Summary Causal Graph, SCG). *Let $\mathcal{G}^{\mathrm{w}} = (\mathbb{E}^{\mathrm{w}}, \mathbb{V}^{\mathrm{w}})$ be a WCG with maximal temporal lag $\gamma$. The underlying summary causal graph (SCG) of $\mathcal{G}^{\mathrm{w}}$ is $\mathcal{G}^{\mathrm{s}} = (\mathbb{E}^{\mathrm{s}}, \mathbb{V}^{\mathrm{s}})$ where $\mathbb{V}^{\mathrm{s}}$ contains one node for each time series and the set of directed edges $\mathbb{E}^{\mathrm{s}}$ are defined as follows: for all $X, Y \in \mathbb{V}^{\mathrm{s}}$, $X \neq Y$, there exists a directed edge from $X$ to $Y$ (denoted as $X \rightarrow Y$) if and only if there exists at least one temporal lag $\ell \geq 0$ such that there exists a directed edge between $X_{t-\ell}$ and $Y_t$ in $\mathcal{G}^{\mathrm{w}}$.*

Note that, unlike the FTCG, the WCG and the ECG, the SCG may contain cycles and in particular self-loops, and two edges in opposite directions, as illustrated in the running example in Figure 3. Unlike the ECG, it is not possible to infer the SCG directly from data without additional assumptions. For example, NBCB$^{\mathrm{acyclic}}$ (Assaad et al., 2021) can discover the summary causal graph assuming it is acyclic, which limits the range of potential WCGs and ECGs.

In this paper, we focus on methods that infer the WCG or the ECG from observational data.

We use throughout for clarity the notation $\mathcal{G}^{\star} = (\mathbb{V}^{\star}, \mathbb{E}^{\star})$ to refer to either $\mathcal{G}^{\mathrm{e}} = (\mathbb{E}^{\mathrm{e}}, \mathbb{V}^{\mathrm{e}})$ or $\mathcal{G}^{\mathrm{w}} = (\mathbb{E}^{\mathrm{w}}, \mathbb{V}^{\mathrm{w}})$. Moreover, we use the notation $t^{\star}$ as a wildcard that represents time step $t - i$, where $i \in \{0, \ldots, \gamma\}$ in the context of WCG, and that represents either $t-$ or $t$ in the context of ECG.

Another important notion that we will use in this paper is the notion of a causal order (Peters et al., 2017) which we define, in the following, only for instantaneous nodes.

**Definition 4** (Causal Order of Instantaneous Nodes). *Given a causal graph $\mathcal{G}^{\star}$, we call a bijective mapping $\pi : \mathbb{V}_t \mapsto \{1, \ldots, d\}$, a causal order of instantaneous nodes if it satisfies*

$$\pi(X_t) < \pi(Y_t) \qquad \text{if } Y_t \in \mathrm{Desc}_{\mathcal{G}^{\star}}(X_t), \forall X_t, Y_t \in \mathbb{V}_t.$$

Because of the acyclicity assumption of the WCG (resp. ECG), there is always a causal order between instantaneous nodes (and a fortiori between all nodes) but it is not necessarily unique (Peters et al., 2017). For example, in Figure 3a, there exists a causal order $\pi_1$ such that $\pi_1(X_t) = 3$ and $\pi_1(W_t) = 4$ and there exists another causal order $\pi_2$ such that $\pi_2(W_t) = 3$ and $\pi_2(X_t) = 4$, while $\pi_1(Y_t) = \pi_2(Y_t) = 1$, $\pi_1(Z_t) = \pi_2(Z_t) = 2$ and $\pi_1(U_t) = \pi_2(U_t) = 5$. Note that the number of possible causal orders is not limited to two, for example, we can obtain a new causal order from $\pi_2$, by inverting the positions of $U_t$ and $X_t$.

## 3 Related works

In this section, we start by giving a general literature review and then we give additional details on the steps that compose noise-based and constraint-based algorithms.

### 3.1 Literature review

In this section, we describe several methods from the main causal discovery families related to our work. More details can be found in thorough reviews such as Assaad et al. (2022a), Hasan et al. (2023), Gong et al. (2023). All those methods rely on consistency throughout time (Assumption 5) to reduce the infinite dimension of the full-time causal graph. Moreover, to consider meaningful graphs, causal sufficiency and causal Markov condition (Assumptions 1 and 2 ) are assumed.

*Granger Causality*, introduced by Granger in 1969 and improved in subsequent works (Granger, 2004, Arnold et al., 2007), stands as one of the earliest methods designed for identifying causal relationships among time series. This approach primarily considers linear relationships and temporal priorities, operating under the assumption that the past of a cause is both necessary and sufficient for optimally forecasting its effect. It is important to note that Granger causality, by relying on temporal priority, is limited in its ability to infer instantaneous causal relations. Furthermore, Granger causality methods typically utilize the past of one time series, up to a defined maximal temporal lag, to predict the present value of another time series without explicitly distinguishing between the importance of different lags. Consequently, these methods can be employed in constructing an ECG if we assume that there are no instantaneous relations.

*Constraint-based approaches* (Spirtes et al., 2000) are certainly the most popular approaches for discovering causal graphs. These methods are based on conditional independence tests, do not depend on any specific distribution form, and require the faithfulness assumption. In the case of causal sufficiency (Assumption 1), they are usually based on the PC-algorithm (Spirtes et al., 2000), initially introduced for non-temporal data. In theory, a constraint-based algorithm can only infer a representative (known as a CPDAG, Chickering, 2002a) of the Markov equivalence class (Verma & Pearl, 1990). Fast Causal Inference (FCI) algorithm can be used when Assumption 1 is not satisfied and infers a Partially Ancestral Graph (PAG), which is a representative of a class of equivalent Maximal ancestral graphs (MAG) and allows representing the existence of hidden confounders in the causal graph. For time series, several algorithms infer WCGs, such as PCMCI (Runge et al., 2019) and PCMCI$^+$ (Runge, 2020), extensions of the PC-algorithm, and tsFCI (Entner & Hoyer, 2010) or SVAR-FCI (Malinsky & Spirtes, 2018), extensions of the FCI algorithm. PCGCE proposed by Assaad et al. (2022c), is based on the adaptation of the PC-algorithm and infers directly an ECG.

*Score-based approaches* (Chickering, 2002a) search over the space of possible graphs, trying to maximize a score that reflects how well the graph fits the data. Greedy Equivalence Search (GES, Chickering, 2002a) is one of the first score-based methods that, under the faithfulness assumption, finds the CPDAG similarly to the PC-algorithm. There have been several recent modifications of GES, such as the more efficient Fast Greedy Search (FGS, Ramsey, 2015) and Selective Greedy Equivalence Search (Chickering & Meek, 2015).

*Continuous optimization-based approches* Unlike score-based methods, this category of algorithms avoids the combinatorial greedy search over DAGs by employing gradient-based optimization. Notears (Zheng et al., 2018) is considered as the first approach to reformulate the greedy search as a continuous optimization problem within this category. This method assumes linearity and considers only DAGs whose topological order follows increasing marginal variance (rescaling data can change or reverse their inferences) (Kaiser & Sipos, 2021). Recently, a new extension of Notears called Dynotears (Pamfil et al., 2020) was presented to infer a WCG from time series.

*Noise-based approaches* discover causal relations using footprints produced by the causal asymmetry in the data, namely the noise. They assume an identifiable functional model, as introduced in Assumption 4. For time series, one of the most popular algorithms in this family is VarLiNGAM (Hyvärinen et al., 2008), which can discover the WCG assuming linear autoregressive models with non-Gaussian noise. TiMINo (Peters et al., 2013) discovers the SCG assuming a nonlinear additive noise model. Note that TiMINo only assumes non-Gaussian noise when causal relations are linear. These approaches also require the minimality condition (Assumption 3).

*Hybrid frameworks* integrate methods from different families to improve the inference of the graph by mitigating the limitations of one algorithm through its combination with another algorithm. For nontemporal data, one group of methods combines constraint-based and score-based methods. Greedy Fast Causal inference (GFCI, Ogarrio et al., 2016) is a combination of the constraint-based FCI algorithm and the GES algorithm,

which allows addressing FCI limited applicability on small sample size data and correcting the graph inferred by GES in the presence of latent confounders (removing causal sufficiency, stated in Assumption 1). HCM hybrid (Li et al., 2022) aims to discover the causal structure from mixed nontemporal data and combines part of the constraint-based method and greedy search adapted for mixed data. SVAR-GFCI introduced by Malinsky & Spirtes (2018) extends GFCI for temporal data. Another type of hybrid method is a combination of constraint-based and noise-based methods. Hoyer et al. (2008) presented a hybrid approach for nontemporal called PClingam that starts with a constraint-based procedure to find the pattern of the graph and then uses a noise-based procedure to orient some of the non-oriented edges (depending on the noise distribution), which allows obtaining a more informative causal graph. Another algorithm called FRITL (Chen et al., 2021) aims to infer more informative causal graphs with the presence of hidden confounders using the noise-based LiNGAM methods to refine the output of the FCI algorithm. For time series, a combination of noise-based and constraint-based methods was proposed by Assaad et al. (2021)—to infer an SCG assuming there are no cycles between different time series—which starts with inferring causal order using an additive noise model and pruning unnecessary edges using conditional independence tests. The combination of methods from different families generally relies on the assumptions required by employed methods (as for example in the case of GFCI or FRTIL), while sometimes hybridization allows for relaxation of some assumptions, e.g., faithfulness in the case of NBCB$^{acyclic}$ or non-Gaussian noise in the case of PClingam.

## 3.2 Details on noise-based and constraint-based methods

Let us first describe the basic steps of noise-based methods. A noise-based algorithm that infers a causal graph relies on the fact that, under an identifiable functional model (Assumption 4), a prediction model of a target node $Y$ where the predictors are the true causes of $Y$ should yield residuals (that represent the noise) that are independent of the causes. The procedure of such an algorithm can be divided into two main steps:

NB1. Find the causal order between instantaneous nodes $\mathbb{V}_t$ by recursively performing regression and independence tests between the predictors and residuals (noise). But note that even if only instantaneous nodes $\mathbb{V}_t$ are accounted for, lagged nodes are used in the regression as a means to account for confounder bias. For example, assuming linearity, we use the following regression model to compute the residuals of each $Y_t \in \mathbb{V}_t$:

$$Y_t = \sum_{X_t \in \mathbb{V}_t \setminus \{Y_t\}} a_{xy} X_t + \sum_{Z_{t-\ell} \in \mathbb{V}^\star \setminus \mathbb{V}_t} a_{zy\ell} Z_{t-\ell} + \xi_t^y. \tag{2}$$

NB2. Find which set of predictors is not needed to keep the independence between other predictors and residuals. The former set of predictors is considered as not causally related to the node that is predicted and the latter set of predictors is considered as the causes of this node.

In general, step NB1 requires only Assumptions 1, 2, 4 and 5. Note that it has been shown that the causal order can be identified even if Assumption 3 is violated (Peters et al., 2014)[2]. However, Peters et al. (2014) only considered the identifiability of the causal order in case of violation of Assumption 3 up to the additive noise model case. However, Assumption 3 is required for step NB2. The output of step NB1 is a causal order $\hat{\pi}$. For step NB1 one can use, for example, TiMINo (Peters et al., 2013) or VarLiNGAM (Hyvärinen et al., 2008) algorithms.

Turning to constraint-based methods, we focus in this paper on the methods that are based on the PC-algorithm, which requires causal sufficiency (Assumption 1). It follows that most constraint-based algorithms for WCGs or ECGs can be divided into two main steps:

CB1. Initialize a fully connected graph such that lagged relations are oriented using temporal priority (if $X_{t^\star}$ is adjacent to $Y_t$ and $t^\star < t$ then $X_{t^\star} \to Y_t$) and instantaneous relations are unoriented, and then prune unnecessary edges using the following procedure:

---

[2]Note that Peters et al. (2014) considered the case where the minimality assumption is violated which is equivalent to the case where Assumption 3 is violated (Peters et al., 2017).

(a) Eliminate edges between nodes $(X_{t^\star}, Y_t)$ if $X_{t^\star} \perp\!\!\!\perp Y_t$.

(b) For each pair of nodes $(X_{t^\star}, Y_t)$ having an edge between them, and for each subset of nodes $\mathbb{S} \subseteq \mathbb{V}^\star \backslash \{X_{t^\star}, Y_t\}$ of size $n = 1$ such that $\forall Z_{t'} \in \mathbb{S}$, $Z_{t'}$ is *adjacent* to $Y_t$, eliminate the edge between $X_{t^\star}$ and $Y_t$ if $X_{t^\star} \perp\!\!\!\perp Y_t \mid \mathbb{S}$.

(c) Iteratively repeat step (b) while increasing the size of the conditioning set $n$ by 1 until there are no more adjacent pairs $(X_{t^\star}, Y_t)$, such that there is a subset $\mathbb{S} \subseteq \mathbb{V}^\star \backslash \{X_{t^\star}, Y_t\}$ of size $n$ that was not tested.

CB2. Orient instantaneous relations using some rules (in case of causal sufficiency, see Spirtes et al., 2000, Meek, 1995).

Note that for step CB1, we only need Assumptions 1, 2, 3 and 5, while steps CB1 and CB2 together require the faithfulness assumption, which is stronger than Assumptions 3 (Ramsey et al., 2006). The output of the CB1 step is a partially oriented graph $\hat{\mathcal{G}}^\star$ that shares the same skeleton as the true graph $\mathcal{G}^\star$ and where all lagged relations are oriented and all instantaneous relations are non-oriented.

In our work, we consider two algorithms from this family. The first one, PCMCI$^+$ (Runge, 2020), is an adaptation of the PC-algorithm to time series that can discover WCGs, and the second one, PCGCE (Assaad et al., 2022c), is an adaptation of the PC-algorithm that can discover ECGs.

## 4 Hybrids of constraint-based and noise-based algorithms

Here, we present a hybrid framework for discovering causal graphs from observational temporal data, which contains two classes of methods: noise-based-then-constraint-based (NBCB) and constraint-based-then-noise-based (CBNB). Both classes of methods operate under identical assumptions. They both necessitate the standard Assumptions 1 and 2 in addition to Assumption 5 for stationary time series. Moreover, both classes require Assumption 4 which is essential for noise-based methods. As both classes use only part of the algorithms from the constraint-based family, they do not require the faithfulness assumption but only the weaker adjacency faithfulness, stated in Assumption 3. In the rest of the paper, we assume that methods from CBNB or NBCB classes are built using the algorithms from constraint-based and noise-based families that are correct under the given assumptions.

Note that we do not need an acyclicity assumption of the true SCG, as in NBCB$^{\text{acyclic}}$, since we primarily focus on inferring a WCG or an ECG. If needed, we can then deduce the SCG from the inferred WCG or the inferred ECG.

In the following sections, we provide a detailed description of the NBCB and CBNB classes of methods.

### 4.1 NBCB class of algorithms

Each of the methods in the NBCB class has two major steps. The first part of NBCB constructs a fully connected graph such that lagged relations are oriented using temporal priority, on which the NB1 step is applied in order to find the causal order between all instantaneous nodes.

In the second part, we aim to use CB1 to prune the edges. This step can be optimized by using additional information on the causal order. We modify CB1 step to CB1$'$, by taking into account the causal order of the nodes in the following way:

CB1$'$. Start with the initialization of the fully connected oriented graph as instantaneous relations can be oriented using the causal order $\pi$.

(a) Eliminate edges between nodes[3] $(X_{t^\star}, Y_t)$ if $X_{t^\star} \perp\!\!\!\perp Y_t$.

(b) For each pair of nodes $(X_{t^\star}, Y_t)$ having an edge between them, and for each subset of nodes $\mathbb{S} \subseteq \mathbb{V}^\star \backslash \{X_{t^\star}, Y_t\}$ of size $n = 1$ such that $\forall Z_{t'} \in \mathbb{S}$, $Z_{t'}$ is *parent* of $Y_t$, eliminate the edge between $X_{t^\star}$ and $Y_t$ if $X_{t^\star} \perp\!\!\!\perp Y_t \mid \mathbb{S}$.

---

[3]We recall that $t^\star$ is a wildcard that represent time step $t - i$, where $i \in \{0, \dots, \gamma\}$ in the context of WCG and represent either $t-$ or $t$ in context of ECG.

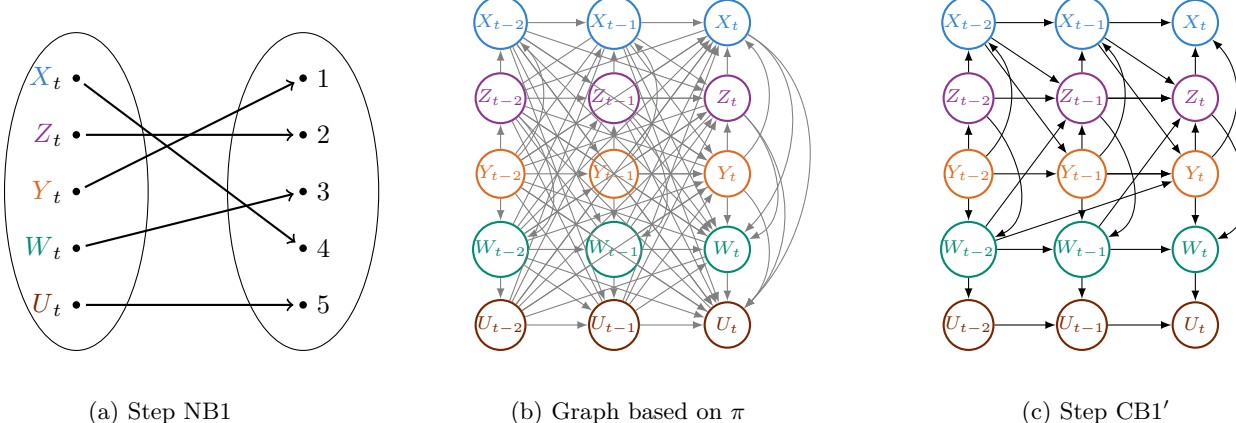

(a) Step NB1            (b) Graph based on $\pi$            (c) Step CB1$'$

Figure 4: Illustration of the NBCB algorithm for the running example. (a) Output of the NB1 step (b) Output of the CB1$'$.

    (c) Iteratively repeat step (b) while increasing the size of the conditioning set $n$ by 1 until there are no more adjacent pairs $(X_{t^\star}, Y_t)$, such that there is a subset $\mathbb{S} \subseteq \mathbb{V}^\star \backslash \{X_{t^\star}, Y_t\}$ of size $n$ that was not tested.

Note that in CB1$'$ step (a) and (c) are the same as in step CB1. The NBCB class of methods can be presented as a combination of the NB1 and CB1$'$, for which we provide a schematic illustration in Figure 4.

**Theorem 1.** *Let $\mathcal{G}^f = (\mathbb{V}^f, \mathbb{E}^f)$ be an FTCG. Under Assumptions 1, 2, 3, 4, 5 and given perfect conditional independence information about all pairs of variables in $\mathbb{V}^f$, any algorithm in the NBCB class returns the correct WCG or the correct ECG compatible with $\mathcal{G}^f$.*

The proof of Theorem 1 is available in Appendix A. It is important to note that after having obtained the WCG or the ECG, we can deduce the correct SCG using Definition 3.

The NBCB class is robust to the violation of Assumption 3. This means that even if Assumption 3 is violated, NBCB is still capable of providing valuable information on causal relationships, as demonstrated in the following proposition. We restrict ourselves to a class of functional identifiable models known as restrictive additive noise models class (Peters et al., 2014), which is stronger than Assumption 4.

**Proposition 1** (Violation of Assumption 3). *Under Assumptions 1, 2, 5, given that the SCM is a restrictive additive noise model (Peters et al., 2014), and given a correct causal order between instantaneous nodes, the NBCB class would give a WCG or an ECG such that, for each pair of nodes $X_{t^\star}$ and $Y_t$, one of the following possibilities holds true:*

    *(1) The causal relationship between $X_{t^\star}$ and $Y_t$ is correctly identified.*

    *(2) $X_{t^\star}$ and $Y_t$ are not adjacent in the inferred graph, but they are adjacent in the true graph.*

    *(3) $X_{t^\star}$ and $Y_t$ are adjacent in the inferred graph, but they are not adjacent in the true graph.*

The proof of Proposition 1 is presented in Appendix A. Proposition 1 states that if there is an oriented edge in a graph $\hat{\mathcal{G}}^\star$ inferred by NBCB, then it can not have an opposite orientation in the true graph. In the following, we illustrate cases (2) and (3) of the Proposition 1. Let us consider that there exists linear dynamic SCM with the corresponding true WCG $\mathcal{G}^w$ presented in Figure 5a. As long as adjacency faithfulness is violated, we can assume that there exists a path $\langle Z_{t-2}, Z_{t-1}, Z_t, X_t \rangle$ which is canceled by the path $\langle Z_{t-2}, X_t \rangle$ (see example of faithfulness violation in Section 2.1), such that in the observed data $Z_{t-2} \perp\!\!\!\perp X_t \mid X_{t-1}$. Thus, the edge $Z_{t-2} - X_t$ would be removed, so we obtain case (2) in Proposition 1. A similar argument works for the edge $Z_{t-2} - Y_t$. Further, these two errors would propagate in the next step for the pair of nodes $Y_t$ and $X_t$. As in the inferred graph $\hat{\mathcal{G}}^w$, edges $Z_{t-2} - X_t$ and $Z_{t-2} - Y_t$ are absent

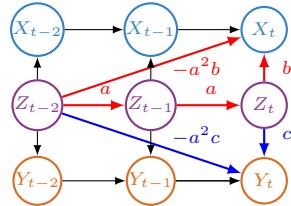

(a) WCG of an SCM violating adjacency faithfulness

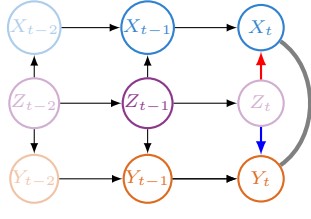

(b) Output of NBCB

Figure 5: Illustration of items (2) and (3) in Proposition 1.

(see Figure 5b), then $X_t \not\perp\!\!\!\perp Y_t \mid \text{Pa}_{\hat{\mathcal{G}}^w}(X_t) \cup \text{Pa}_{\hat{\mathcal{G}}^w}(Y_t)$, as $Z_{t-2}$ is the common confounder in the true graph, thus the edge $X_t - Y_t$ would not be removed and we obtain the case (3) in Proposition 1.

We want to highlight that Proposition 1 holds when the CB1′ step of NBCB is implemented using a PC-style algorithm. Other constraint-based algorithms could be envisioned, such as a greedy approach where the iterative process of finding conditional independence while progressively expanding the conditioning set (steps (b) and (c) of CB1′) is replaced by a single step of conditioning on all parents. With such a greedy algorithm, the concerns outlined in case (3) of the proposition could be alleviated.

Note that NBCB would perform poorly when the SCM is not an identifiable functional model (Assumption 4 is violated). In this case, NB1 would give an incorrect causal order, and the errors would propagate in the second step.

---

**Algorithm 1:** Noise-Based-then-Constraint-Based (NBCB)

---

**Input:** A multivariate time series, a maximal temporal lag $\gamma$ and a significance threshold $\alpha$, NB1,
     CB1′, an independence measure I(), and a conditional independence test CI()

**Result:** $\hat{\mathcal{G}}^\star$ (WCG or ECG) and $\hat{\mathcal{G}}^s$ (SCG)

Find the causal order $\hat{\pi}$ between all instantaneous nodes using NB1 which takes $\gamma$ and I() as
  hyper-parameters;

Discover $\hat{\mathcal{G}}^\star$ using CB1′ which takes $\gamma$, $\alpha$, CI(), and $\hat{\pi}$ as hyper-parameters;

Deduce the SCG $\hat{\mathcal{G}}^s$ from $\hat{\mathcal{G}}^\star$ using Definition 3.

---

The pseudo-code of NBCB is given in Algorithm 1, which involves the abstract steps NB1 and CB1′. Step NB1 can be directly obtained from any existing noise-based algorithm. In the experimental section of this paper, step NB1 is directly based on the VarLiNGAM algorithm, while step CB1′ is either based on the modification of the PCTMI$^+$ algorithm or on the modification of the PCGCE algorithm. More details on specific versions of NB1 and CB1′ is given in Appendix B.

## 4.2 CBNB class of algorithms

As far as we know, there exists no previous work that investigated the case where the constraint-based part of a hybrid method is executed before the noise-based part in causal discovery from time series (for non-temporal data, see PClingam, Hoyer et al., 2008). In theory, there is no clear argument as to why one part should be executed before the other. So here, we present a new class of methods called CBNB, where the constraint-based part is before the noise-based part.

In the first step of CBNB, CB1 is used to infer $\hat{\mathcal{G}}^\star$. Then, we are going to orient edges between instantaneous nodes $\mathbb{V}_t$ by finding the causal order between them.

A vanilla approach would be to apply NB1 over all instantaneous nodes (while taking into account lagged common confounders), as it was done in NBCB, to get the causal order and then orient the instantaneous relations accordingly. However, we argue that this vanilla approach, despite being correct, is not optimal since it does not take into account the knowledge that has already been acquired through the construction

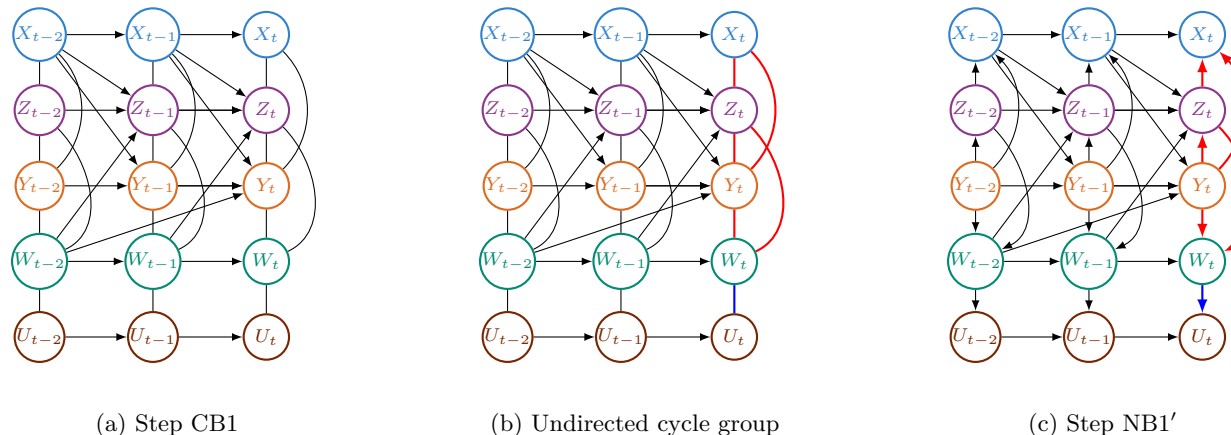

| (a) Step CB1 | (b) Undirected cycle group | (c) Step NB1′ |

Figure 6: Illustration of the CBNB algorithm for the running example. (a) Output of the CB1 step (b) Detection of the undirected cycle group, where red and blue colours denote two different cycle groups of instantaneous nodes (c) Output of the NB1′ on each of the cycle groups.

of the output graph in step CB1. Therefore, we modify the NB1 step to NB1′ by finding the causal order within different groups of instantaneous nodes separately in the following way:

NB1′. Find the causal order within instantaneous groups of nodes $\mathbb{I}_t \subseteq \mathbb{V}_t$ by recursively performing regression and independence tests between the predictors and residuals (noise). But note that even if only the instantaneous group of nodes $\mathbb{I}_t$ are accounted for, the lagged parents $Pa_{\hat{\mathcal{G}}^\star}(\mathbb{I}_t)\backslash\mathbb{V}_t$ are used in the regression as a means to account for confounder bias.

Not any groups of instantaneous nodes $\mathbb{I}_t$ would yield a correct causal order since, in general, for any $X_t, Y_t \in \mathbb{I}_t$, $\mathbb{I}_t \cup Pa_{\hat{\mathcal{G}}^\star}(\mathbb{I}_t)$ does not necessarily contain any subset that remove confounding bias between $X_t$ and $Y_t$. Therefore, these groups should be selected carefully. To find these groups, we first need to define an undirected cycle walk.

**Definition 5** (Undirected Cycle Walk)**.** *Let $\hat{\mathcal{G}}^\star$ be the output of the CB1 step. A sequence of edges $\boldsymbol{u} = \langle e_1, \ldots, e_n \rangle$, for $n \geq 2$, is an* undirected cycle walk *iff*

- *$\forall e_i \in \boldsymbol{u}$, $e_i$ is an undirected edge in $\hat{\mathcal{G}}^\star$;*

- *$\forall e_i, e_j \in \boldsymbol{u}$, such that $i < j$, no node in $e_i$ coincides with any node in $e_j$, except if $j = i+1$ where the right-hand side node in $e_i$ always coincide with the left-hand side node in $e_{i+1}$, or if $i = 1$, $j = n$, where the left-hand side node in $e_1$ and the right-hand side node in $e_n$ always coincide.*

Note that in our definition an undirected edge can form an undirected cycle walk. We provide a schematic illustration in Figure 6 for the running example. In Figure 6b, $\langle X_t - Y_t, Y_t - Z_t, Z_t - X_t \rangle$ and $\langle Z_t - Y_t, Y_t - W_t, W_t - Z_t \rangle$ are two undirected cycle walks. Notice that $Z_t - Y_t$ is in both undirected cycle walks and would be considered twice, which is not only a computational problem but might also induce bias in practice. Thus, defining a undirected cycle walk alone is not sufficient. So, we combine all undirected cycle walks that share at least one edge in a undirected cycle group[4] which is defined as follows:

**Definition 6** (Undirected Cycle Group)**.** *Let $\hat{\mathcal{G}}^\star$ be the output of the CB1 step. $\mathbb{C}$ is an* undirected cycle group *of $\hat{\mathcal{G}}^\star$ iff $\mathbb{C}$ is a set of undirected cycle walks and $\forall \boldsymbol{u}_1, \boldsymbol{u}_2 \in \mathbb{C}$, $\boldsymbol{u}_1 \cap \boldsymbol{u}_2 \neq \emptyset$.*

We assume that two undirected cycle walks $\mathbf{u}_1$ and $\mathbf{u}_2$ do not intersect (i.e., $\mathbf{u}_1 \cap \mathbf{u}_2 = \emptyset$) if there is no common edge between these two walks. For simplicity, we consider two undirected cycle walks that are reversed versions of one another as equivalent and so only one of these two undirected cycle walks will be

---

[4]Note that our definition of cycle group is different than the one introduced in Spirtes (1995).

accounted for. In Figure 6b, for the WCG, there are two *undirected cycle groups*, $\mathbb{C}_1 = \{\langle X_t - Y_t, Y_t - Z_t, Z_t - X_t \rangle, \langle Z_t - Y_t, Y_t - W_t, W_t - Z_t \rangle, \cdots\}$ and $\mathbb{C}_2 = \{\langle U_t - W_t, W_t - U_t \rangle\}$. The same example applies to the corresponding ECG. Given an *undirected cycle group* $\mathbb{C}$, we are interested in the set of its nodes, meaning the set of nodes that belong to undirected cycle walks described by the undirected cycle group. Similarly, we say that an edge $X_t - Y_{t'}$ belongs to $\mathbb{C}$ iff $\exists \mathbf{u} \in \mathbb{C}$ such that $\langle X_t, Y_{t'} \rangle \in \mathbf{u}$.

Having brought out the concept of *undirected cycle group*, we can now describe the second (noise-based) part of CBNB. Given the output of the CB1 step, CBNB searches for all undirected cycle groups. Note that each edge is a member of exactly one undirected cycle group. Then for each undirected cycle group $\mathbb{C}$, CBNB uses NB1′ to find the causal order $\pi$ between all the nodes belonging to undirected cycle group $\mathbb{C}$. Finally, using this causal order, CBNB orient all edges belonging to $\mathbb{C}$.

Note that the CBNB step where NB1′ is applied for different *undirected cycle groups* can be parallelized, which can significantly improve the computational time for high dimensional problems.

**Theorem 2.** *Let $\mathcal{G}^f = (\mathbb{V}^f, \mathbb{E}^f)$ be an FTCG. Under Assumptions 1, 2, 3, 4, 5 and given perfect conditional independence information about all pairs of variables in $\mathbb{V}^f$, any algorithm in the CBNB class returns the correct WCG or the correct ECG compatible with $\mathcal{G}^f$.*

The proof of Theorem 2 is available in Appendix A. It is also important to note that as in the case of NBCB, after having obtained the WCG or the ECG, we can deduce the correct SCG using Definition 3.

The CBNB class is robust to the violation of Assumption 4. This means that even if Assumption 4 is violated, CBNB is still capable of providing valuable information on causal relationships, as demonstrated in the following proposition.

**Proposition 2** (Violation of Assumption 4). *Under Assumptions 1, 2, 3, 5 and given perfect conditional independence information about all pairs of variables, CBNB is guaranteed to find the correct skeleton of the WCG or the ECG.*

The proof of Proposition 2 is available in Appendix A. Note that since CBNB gives the correct skeleton of WCGs and ECGs, the correct skeleton of SCG can also be deduced. If Assumption 3 is violated, then the skeleton obtained in the CB1 step of CBNB is not reliable, thereby affecting the reliability of the CBNB algorithm's results.

The pseudo-code of CBNB is given in Algorithm 2, which involves the abstract steps CB1 and NB1′. In the experimental section of this paper, step CB1 is directly based either on the PCMCI$^+$ algorithm or on the PCGCE algorithm, while step NB1′ is based on the modification of the VarLiNGAM algorithm. More details on specific versions of CB1 and NB1′ is given in Appendix B.

---

**Algorithm 2:** Constraint-Based-then-Noise-Based (CBNB)

---

**Input:** A multivariate time series, a maximal temporal lag $\gamma$ and a significance threshold $\alpha$, CB1, NB1′, an independence measure I(), and a conditional independence test CI()
**Result:** $\hat{\mathcal{G}}^\star$ (WCG or ECG) and $\hat{\mathcal{G}}^s$ (SCG)
Initialize $\hat{\mathcal{G}}^\star$ as the output of CB1 which takes $\gamma$, $\alpha$, and CI() as hyper-parameters;
**for** *each undirected cycle group $\mathbb{C}$ in $\hat{\mathcal{G}}^\star$* **do**
> $\mathbb{I}_t$: set of nodes that belong to $\mathbb{C}$;
> Find the causal order $\hat{\pi}$ between nodes in $\mathbb{I}_t$ using NB1′ which takes $\gamma$, I(), and $\hat{\mathcal{G}}^\star$ as hyper-parameters;
> Orient instantaneous edges between $\mathbb{I}_t$ in $\hat{\mathcal{G}}^\star$ using the order $\hat{\pi}$;

Deduce the SCG $\hat{\mathcal{G}}^s$ from $\hat{\mathcal{G}}^\star$ using Definition 3

---

### 4.3 Complexity analysis for NBCB and CBNB classes of methods

When considering $d$ time series, finding the causal order is of complexity $d^2 f(n, d)$ where $f(n, d)$ is the complexity of the user-specific regression method with time series length $n$. Then, pruning the graph by

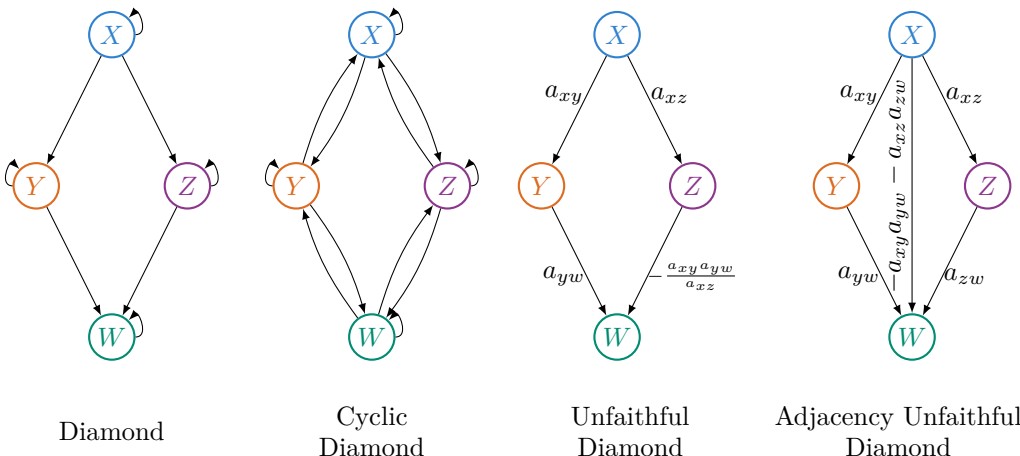

Figure 7: Summary causal graphs corresponding to the data simulated in Section 5.2. The last two models correspond to an unfaithful distribution.

conditional independence tests is of complexity $\frac{(dq)^2(dq-1)^{k-1}}{(k-1)!}$, where $k$ represents the maximal degree of any node, $q$ is equal to $\tau$ when considering a WCG and $q$ is equal to 2 when considering an ECG, and each operation consists in conducting significance test to a conditional independence measure. Both CBNB and NBCB have the same complexity in the worst case, $d^2 f(n,d) + \frac{(dq)^2(dq-1)^{k-1}}{(k-1)!}$ which corresponds for CBNB to the case where there is only one undirected cycle group.

# 5 Experiments

In this section, we propose first an extensive analysis[5] both on simulated data, generated from basic causal structures, and on simulated but realistic benchmarks. We then perform an analysis on different real datasets.

## 5.1 Experimental setup

**Baselines and hyper-parameters.** We compare NBCB and CBNB with five state-of-the-art methods:

- the multivariate version of Granger Causality denoted GCMVL (Arnold et al., 2007);

- the constraint-based methods PCMCI$^+$ (Runge, 2020) for which we use the Python code available at https://github.com/jakobrunge/tigramite and PCGCE (Assaad et al., 2022c) for which the main Python code available at https://github.com/ckassad/PCGCE. For PCGCE, as the authors suggested, we reduce the dimensionality of $\mathbb{V}_{t-}$ in the ECG to 1 using PCA;

- the continuous optimization-based method Dynotears (Pamfil et al., 2020), for which the Python code is available at https://github.com/quantumblacklabs/causalnex. For this method, we set the hyperparameters to their recommended values ($\lambda_W = \lambda_A = 0.05$ and $\alpha_W = \alpha_A = 0.01$);

- the noise-based method VarLiNGAM (Hyvärinen et al., 2008) for which we use the Python code available at https://github.com/cdt15/lingam where the regularization parameter in the adaptive Lasso is selected using BIC.

For all the methods, the maximal temporal lag is set to $\gamma = 5$ and the significant threshold for hypothesis testing to $\alpha = 0.05$. We test two versions of each of our classes which we denote NBCB-w, NBCB-e, CBNB-w, and CBNB-e. Methods with "-w" suffix are based on the algorithms that infer a WCG, while methods

---

[5]A Python code of all our methods and of every experimentation is available in https://github.com/ckassaad/Hybrids_of_CB_and_NB_for_Time_Series.

with "-e" are based on the algorithms that infer an ECG. In NBCB-w and NBCB-e, the NB1 step is based on the VarLiNGAM algorithm and the CB1' step is respectively based on the PCMCI$^+$ algorithm and the PCGCE algorithm. In CBNB-w and CBNB-e, the CB1 step is respectively based on the PCMCI$^+$ algorithm and the PCGCE algorithm and the NB1' step is based on the VarLiNGAM algorithm. The pseudo-code of each version of the NB1, CB1, NB1', CB1' steps are given in Appendix B. To find undirected cycle walks in CBNB-w and CBNB-e, we use an adapted version of Paton's algorithm (Paton, 1969). For all methods that require a conditional independence test, we use a test based on partial correlation, which assumes Gaussian distributions but which has been successfully used on non-iid data (Peters et al., 2013).

For real data, we also use nonlinear versions of our hybrid algorithms as well of the PCMCI and the PCGCE algorithms. We use kernel-based conditional independence test which combines Gaussian process regression with a distance correlation test on the residuals (Runge et al., 2019), and a Gaussian process regression in the noise-based part. All nonlinear versions are denoted by the suffix "-nl".

**Evaluation.** In the different experimental settings, we compared the results concerning the F1 score of the orientations in the SCG obtained without considering self causes, as it is treated differently depending on the methods. When there are three datasets or more, we report the mean and the variance for the F1 score.

## 5.2 Simulated data with Gaussian and non-Gaussian noise

The simulated datasets correspond to four causally sufficient SCGs presented in Figure 7, extracted from WCGs $\mathcal{G}^w$, among which three are acyclic, two correspond to an unfaithful distribution, and one is cyclic. The generating process of all datasets is the following: for all $Y$, for all $t > 0$,

$$Y_t = a_y Y_{t-1} + \sum_{X_{t-\ell} \in \mathrm{Pa}_{\mathcal{G}^w}(Y_t, \mathcal{G}^w)} a_{xy} X_{t-\ell} + 0.1 \xi_t^y,$$

where $a_y, a_{xy} \in U([-1, -0.1] \cup [0.1, 1])$ and for each parent $X$ we randomly choose if $X$ causes $Y$ instantaneously or with a lag of 1, i.e., $\ell \in \{0, 1\}$. Regarding the noise, we consider two different settings, in the first the noise is drawn from a uniform distribution, i.e., $\xi^y \sim U([-1, 1])$, and in the second the noise is drawn from a Gaussian distribution, i.e., $\xi^y \sim N(0, 1)$. For each setting and each structure in Figure 7 we generate 100 datasets of 1000 timestamps.

For the unfaithful diamond structure in Figure 7, following Zhalama et al. (2016), we set $a_x, a_y, a_z, a_w$ to zero, and $a_{zw} = -a_{xy}a_{yw}/a_{xz}$. All relationships are considered instantaneous. Consequently, in the distribution consistent with this model, we observe $X \perp\!\!\!\perp W$, which violates the faithfulness assumption as this independence is not entailed by the causal Markov condition. However, it does not violate the adjacency faithfulness assumption since $X$ is not adjacent to $W$ in the graph. Similarly, for the adjacency unfaithful diamond structure, we set $a_{xw} = -a_{xy}a_{yw} - a_{xz}a_{zw}$, which violates the adjacency faithfulness assumption since in addition to $X$ being adjacent to $W$ in the graph, in the distribution, we observe $X \perp\!\!\!\perp W$, and this independence is not entailed by the causal Markov condition.

In Table 1, we report the results for the setting with non-Gaussian noise (top panel), and with Gaussian noise (bottom panel).

Let us start with the case of non-Gaussian noise, which is easier to handle for most methods. For the first two structures, diamond and cyclic diamond, all assumptions required by both families of methods are satisfied, as well as by competitors. In this case, VarLiNGAM has the best performance for diamond, followed by CBNB-w and NBCB-w, while CBNB-w and NBCB-w have the best performance for cyclic diamond, followed by VarLiNGAM. PCMCI$^+$ comes closest to these methods. As expected from these results, we can see that CBNB-w and NBCB-w perform better than both original noise-based and constraint-based methods or as a trade-off between them. We can note that the same conclusion can be made for NBCB-e and CBNB-e. For the unfaithful structures, constraint-based methods PCMCI$^+$ and PCGCE have a drop in performance due to faithfulness violation. VarLiNGAM has the best results, which makes sense as it does not rely on faithfulness, followed by CBNB-e, with close results by NBCB-e. We can also note that our methods experience a noticeable drop in performance only for adjacency unfaithful structures, confirming that they

Table 1: Results obtained on the simulated data of Section 5.2 for the different structures with 1000 observations with non-Gaussian noise (top panel) and with Gaussian noise (bottom panel). We report the mean and the variance of the F1 score of the orientations in the SCG. The best results are in blue bold and the second best results are in green bold.

| | Diamond | Cyclic Diamond | Unf. Diamond | Adj. unf. Diamond |
|---|---|---|---|---|
| | | Non-Gaussian noise | | |
| NBCB-w | **0.94** ± 0.01 | **0.81** ± 0.01 | 0.86 ± 0.01 | 0.8 ± 0.01 |
| CBNB-w | **0.94** ± 0.01 | **0.8** ± 0.01 | 0.9 ± 0.01 | 0.84 ± 0.01 |
| NBCB-e | 0.74 ± 0.02 | 0.72 ± 0.01 | 0.95 ± 0.01 | **0.86** ± 0.01 |
| CBNB-e | 0.74 ± 0.02 | 0.7 ± 0.02 | **0.96** ± 0.01 | **0.86** ± 0.01 |
| GCMVL | 0.86 ± 0.01 | 0.68 ± 0.01 | 0.04 ± 0.01 | 0.04 ± 0.01 |
| PCMCI$^+$ | 0.92 ± 0.01 | 0.75 ± 0.01 | 0.47 ± 0.04 | 0.44 ± 0.03 |
| PCGCE | 0.69 ± 0.02 | 0.66 ± 0.01 | 0.5 ± 0.01 | 0.45 ± 0.01 |
| Dynotears | 0.03 ± 0.01 | 0.0 ± 0.0 | 0.0 ± 0.0 | 0.0 ± 0.0 |
| VarLiNGAM | **0.99** ± 0.01 | 0.79 ± 0.01 | **0.98** ± 0.01 | **0.87** ± 0.01 |
| | | Gaussian noise | | |
| NBCB-w | 0.78 ± 0.03 | **0.77** ± 0.01 | **0.52** ± 0.05 | **0.48** ± 0.01 |
| CBNB-w | 0.8 ± 0.04 | **0.75** ± 0.01 | **0.52** ± 0.05 | **0.48** ± 0.06 |
| NBCB-e | 0.64 ± 0.02 | 0.67 ± 0.02 | **0.52** ± 0.07 | **0.44** ± 0.06 |
| CBNB-e | 0.72 ± 0.03 | 0.65 ± 0.02 | **0.53** ± 0.07 | **0.44** ± 0.05 |
| GCMVL | **0.87** ± 0.01 | 0.7 ± 0.01 | 0.03 ± 0.01 | 0.01 ± 0.01 |
| PCMCI$^+$ | **0.93** ± 0.01 | **0.75** ± 0.01 | 0.42 ± 0.05 | 0.4 ± 0.04 |
| PCGCE | 0.69 ± 0.02 | 0.65 ± 0.01 | 0.5 ± 0.02 | **0.44** ± 0.01 |
| Dynotears | 0.06 ± 0.02 | 0.0 ± 0.0 | 0.0 ± 0.0 | 0.0 ± 0.0 |
| VarLiNGAM | 0.78 ± 0.03 | 0.74 ± 0.01 | 0.5 ± 0.07 | 0.42 ± 0.06 |

do not require a full faithfulness assumption but are surprisingly still competitive for adjacency unfaithful structure, illustrating some robustness.

Considering Gaussian noise is more challenging, as the SCM is not an identifiable functional model (Assumption 4 is violated), which is needed for VarLiNGAM (and its use within the proposed methods). For the first diamond structure, as expected, PCMCI$^+$ performs best, closely followed by GCMVL and CBNB-w, NBCB-w, and VarLiNGAM. NBCB-e and CBNB-e demonstrate lower performance than VarLiNGAM, due to lower results of PCCGE. For the cyclic diamond structure, NBCB-w has the best results, followed by CBNB-w and PCMCI$^+$. For unfaithful structures with Gaussian noise, the constraint-based methods PCMCI$^+$ and PCGCE again experience a drop in performance due to faithfulness violation. Our methods yield the best results. Specifically, CBNB-e performs best for the unfaithful diamond, with the rest of our methods being the second-best. NBCB-w and CBNB-w work best for the adjacency unfaithful diamond, followed by NBCB-e, CBNB-e algorithms, and PCGCE.

We highlight the consistently poor performance of Dynotears in Table 1, where this method has the lowest result in all scenarios. This can be attributed to the fact that in our simulated data, the variances do not increase in accordance with the topological order of the WCG.

Comparing PCMCI$^+$ and PCGCE, we can see that PCGCE has lower performance in general, except for the unfaithful cases for both types of noise distribution. This empirical observation suggests that PCGCE is more robust to assumption violation. This behavior is also inherited by CBNB-e and NBCB-e methods.

We can conclude that when necessary assumptions are satisfied, CBNB-w and NBCB-w are either trade-offs between the PCMCI$^+$ and VarLiNGAM or perform better, and NBCB-e and CBNB-e, are trade-offs between PCGCE and VarLiNGAM. For the unfaithful structures under assumption violation, all our methods are more robust compared to constraint and noise-based families.

Table 2: Results for realistic datasets of Section 5.3 generated using the Lotka–Volterra model with five species (left column) and ten species (right column). We report the mean and the variance of the F1 score of the orientations in the SCG. The best results are in blue bold and the second best results are in bold green.

| | Lotka–Volterra(5) | Lotka–Volterra(10) |
|---|---|---|
| NBCB-w | $0.41 \pm 0.03$ | $\mathbf{0.28} \pm 0.01$ |
| CBNB-w | $0.38 \pm 0.03$ | $\mathbf{0.24} \pm 0.01$ |
| NBCB-e | $\mathbf{0.47} \pm 0.02$ | $\mathbf{0.24} \pm 0.01$ |
| CBNB-e | $\mathbf{0.44} \pm 0.02$ | $0.23 \pm 0.01$ |
| GCMVL | $0.19 \pm 0.03$ | $0.11 \pm 0.01$ |
| PCMCI$^+$ | $0.36 \pm 0.03$ | $0.22 \pm 0.01$ |
| PCGCE | $\mathbf{0.44} \pm 0.02$ | $0.22 \pm 0.01$ |
| Dynotears | $0.18 \pm 0.05$ | $0.15 \pm 0.01$ |
| VarLiNGAM | $0.43 \pm 0.06$ | $0.23 \pm 0.02$ |

Table 1 presents the F1 score of the orientations in the SCG, which is not suitable to illustrate the Proposition 2. Thus, in Table 6 in Appendix E we present the F1 score on the adjacencies, which illustrate the robustness of CBNB class to Assumption 4. More precisely, CBNB-w has the same performance as PCMCI$^+$ and better than the results of NBCB-w, in the case of Gaussian noise, for all structures, except the adjacency unfaithful diamond. We also see similar results for CBNB-e. Proposition 1 is difficult to see from Table 1, as we do not evaluate separately orientations and adjacencies, due to the specific structure of the inferred graph.

### 5.3 Realistic ecological data from the Lotka–Volterra model

We consider the simulation with multi-species generalization of the Ricker model introduced by Poggiato et al. (2022), which is analogous to the generalized Lotka–Volterra model with abiotic control presented in the same paper and is commonly used in ecological studies. Ricker model with abiotic control in discrete time for abundance of species $Y$ at time step $t$ has the following form:

$$Y_t = \begin{cases} Y_{t-1} \exp\left(\Delta t \Big(\sum_{X_{t-1} \in \mathrm{Pa}_{\mathcal{G}^w}(Y_t, G^w)} a_{xy}X_{t-1} + \overline{Y}(-a_y)\exp\left(-\frac{(o_y - x)^2}{2\sigma_y^2}\right)\Big)\right) + \xi_t^y, & \text{(preys)} \\ Y_{t-1} \exp\left(\Delta t \Big(\sum_{X_{t-1} \in \mathrm{Pa}_{\mathcal{G}^w}(Y_t, G^w)} a_{xy}X_{t-1} - \mu\Big)\right) + \xi_t^y, & \text{(predators)} \end{cases}$$

where $Y_t$ is the abundance of species $Y$ at time $t$, the upper equation is related to preys and the lower to predator species, $\overline{Y}$ is the abundance of species $Y$ in the stationary state, $a_{xy}$ is the strength of the effect of species $X$ on species $Y$ and $a_y$ is strength of the effect of species $Y$ on itself, $\xi_t^y$ is an i.i.d Gaussian random variable with variance $\sigma_r$, $o_y$ is a niche optimum for species $Y$, $x$ is the environmental variable, $\mu$ is the extinction rate of the predator. We run this simulation for the number of species $S = \{5, 10\}$ with the following fixed parameters: fixed environment $x = 0.5$, number of time steps $T = 1000$, $\mu = 0.05$, $\sigma_r = 0.2$, for each species $o_y$ randomly sampled from $U([0.05, 0, 95])$, the interaction matrix related to coefficients $a_{xy}$ and $a_y$ is obtained through a randomly generated WCG $\mathcal{G}^w$ which is compatible with an SCG $\mathcal{G}^s$. The SCG is constrained to contain only bi-directed edges and to encompass precisely 3 trophic levels, representing the hierarchical positions of species in the food chain. Specifically, these levels include basal species or prey $(L_1)$, their predators $(L_2)$, and the predators of predators $(L_3)$. The second constraint ensures that $\forall X \in L_i$ and $\forall Y \in L_j$, if $X \leftrightarrows Y$ in $\mathcal{G}^s$ then $|i - j| = 1$ (for more details see Poggiato et al., 2022). The interaction strength is randomly sampled for all interactions. We generate 100 graphs and thus we obtain 100 datasets.

For the Lotka-Volterra(5) datasets, NBCB-e performs better than the other methods followed by CBNB-e and PCGCE as shown in Table 2. Close to them perform VarLiNGAM, NBCB-w and CBNB-w. Dynotears and GCMVL have the lowest results. For the Lotka-Volterra(10) datasets, all results saw a significant decrease, with NBCB-w performing the best, followed by CBNB-w and NBCB-e. It is worth noting that for all datasets, NBCB-e and CBNB-e perform either better or equally as well as PCGCE and VarLiNGAM, while NBCB-w and CBNB-w outperform PCMCI.

Table 3: Results for real datasets of Section 5.4 using linear methods. We report the mean and the variance (when meaningful, see data description) of the F1 score of the orientations in the SCG. The best results are in blue bold and the second best results are in green bold.

| | Temp. | Veil1 | Veil2 | Dairy | Ingest. | Web1 | Web2 | Antivirus1 | Antivirus2 |
|---|---|---|---|---|---|---|---|---|---|
| NBCB-w | **1** | **1** | 0 | **0.4** | $0.47 \pm 0.03$ | 0.2 | 0.23 | 0.13 | 0.3 |
| CBNB-w | **1** | **1** | 0 | **0.4** | $0.46 \pm 0.11$ | **0.24** | 0.29 | 0.18 | 0.18 |
| NBCB-e | **1** | **1** | 1 | **0.4** | $0.5 \pm 0.05$ | **0.24** | **0.42** | **0.29** | **0.38** |
| CBNB-e | **1** | **1** | 1 | **0.4** | $\mathbf{0.52} \pm 0.06$ | 0.15 | **0.38** | **0.33** | 0.27 |
| GCMVL | 0.67 | **1** | 1 | 0.33 | $0.5 \pm 0.03$ | 0.19 | 0.0 | 0.08 | 0.0 |
| PCMCI$^+$ | **1** | **1** | 0 | **0.4** | $0.3 \pm 0.08$ | 0.17 | 0.32 | 0.04 | 0.11 |
| PCGCE | **1** | **1** | 1 | **0.4** | $\mathbf{0.55} \pm 0.03$ | 0.21 | 0.34 | **0.29** | **0.36** |
| Dynotears | 0.67 | **1** | 1 | 0.33 | $0.25 \pm 0.06$ | 0.22 | 0.3 | 0.18 | 0.17 |
| VarLiNGAM | **1** | **1** | 1 | **0.5** | $0.49 \pm 0.05$ | **0.23** | 0.2 | 0.18 | 0.18 |

## 5.4 Real data

Nine different real datasets are considered in this study. Taking into account the limitations of certain methods in handling nonlinearity, we start by evaluating our algorithms using linear tests and linear regressions, alongside linear tests for constraint-based methods. Then, we proceed to compare the nonlinear counterparts of our methods with those of the constraint-based methods while providing a computational time analysis.

### 5.4.1 The linear case

We detail the performance of each method in the following paragraphs, while the results are summarized in Table 3. See Appendix D for URL links to the considered datasets.

**Temperature.** This is a bivariate time series of length 168 about indoor $I$ and outdoor $O$ measurements. As noted by Assaad et al. (2021), it is expected that $O$ causes $I$.

NBCB-w, CBNB-w, NBCB-e, CBNB-e, PCMCI$^+$, PCGCE, and VarLiNGAM correctly infer $O \rightarrow I$. GCMVL and Dynotears infer a bidirected causal relation.

**Veilleux.** We considered two datasets for Figure 11(a) and 12(a) (Jost & Ellner, 2000) from Veilleux (1979) which study interactions between predatory ciliate Dinidum nasutum and its prey Paramecium aurelia with different values of Cerophyl concentrations (CC): 0.375 and 0.5. The lengths of the time series are 71 and 65. These data were previously analyzed with causal discovery algorithms (Barraquand et al., 2021, Sugihara et al., 2012), which showed bidirectional relationships in both cases.

Here, NBCB-e, CBNB-e, GCMVL, PCGCE, Dynotears and VarLiNGAM discover bidirected relationships Paramecium $\leftrightarrows$ Didinium in both datasets, which is consistent with Sugihara et al. (2012), Barraquand et al. (2021). CBNB-w, NBCB-w, PCMCI$^+$ have detected bidirected relationships between Paramecium and Didinium only in the first dataset.

**Diary.** This dataset provides ten years (from 09/2008 to 12/2018) of monthly prices for milk $M$, butter $B$, and cheddar cheese $C$, so the three time series are of length 124. We expect that the price of milk is a common cause of the price of butter and the price of cheddar cheese: $B \leftarrow M \rightarrow C$.

NBCB-w, CBNB-w, NBCB-e, CBNB-e, PCMCI$^+$, PCGCE and VarLiNGAM correctly inferred that $M \rightarrow B$. But NBCB-w and CBNB-w wrongly inferred that $M \leftarrow C \rightarrow B$, NBCB-e, CBNB-e, wrongly inferred that $M \leftarrow B \leftarrow C$, PCMCI$^+$ and PCGCE wrongly inferred that $M \leftarrow C \leftarrow B$ and VarLiNGAM wrongly inferred that $C \rightarrow B$. GCMVL wrongly infers $B \leftrightarrows M \leftarrow C \rightarrow B$ and Dynotears wrongly infers $M \leftrightarrows B \leftrightarrows C$.

**Ingestion mini.** This benchmark provided by EasyVista consists of 24 datasets each containing three time series with 1000 timestamps collected from an IT monitoring system with a one-minute sampling rate. Half

the datasets are compatible with the graph $C.M.I \leftarrow M.E \rightarrow G.H.I$ (Assaad et al., 2022b) where $M.E$ is the metric extraction which represents the activity of the extraction of the metrics from the messages; $G.H.I$ is the group history insertion, which represents the activity of the insertion of the historical status in the database; and $C.M.I$ is the collector monitoring information, which represents the activity of the updates in a given database. The second half of the datasets are compatible with the graph $M.D \rightarrow M.E \rightarrow M.I$ where $M.D$ is the metric dispatcher which represents the activity of a process that orients messages to other processes with respect to different types of messages; and $M.I$ is the metric insertion which represents the activity of insertion of data in a database. Lags between time series are unknown, as well as the existence of self-causes.

From Table 3, we can see that PCGCE has the best results followed by CBNB-e then by NBCB-e and GCMVL. After that comes VarLiNGAM and NBCB-w and CBNB-w and PCMCI$^+$.

Finally, Dynotears has the worst result. This might suggest that the lags between causes and effects are not consistent over time, in the sense that if the lags between two-time series vary (while respecting the maximal temporal lag), the extended summary causal graph might remain the same however this is not true for the window causal graph. In this case, we might expect that methods inferring window causal graphs (such as PCMCI$^+$) would perform worse than methods inferring extended summary causal graphs (such as PCGCE).

**Web.** We consider a dataset that reflects the activity in a web server which is provided by EasyVista. This dataset contains ten time series collected with a one-minute sampling rate. The raw data of this case study were initially misaligned. In order to align them, we use the two pre-processing strategies described in Appendix C. We denote the dataset pre-processed using Strategy 1 as Web 1 and the dataset pre-processed using Strategy 2 as Web 2. The two processed datasets contain 3000 timestamps. The corresponding summary causal graph for Web dataset is presented in Figure 8a where *NetIn* represents the data received by the network interface card in Kbytes/second; *NetOut* represents the data transmitted out by the network interface card in Kbytes/second; *NPH* represents the number of HTTP processes; *NPP* represents the number of PHP processes; *NCM* represents the number of open MySql connections which are started by PHP processes; *CpuH* represents the percentage of CPU used by all HTTP processes; *RamH* represents the percentage of RAM used by all HTTP processes; *CpuP* represents the percentage of CPU used by all PHP processes; *DiskW* represents the Disk write in Kbytes/second; *CpuG* represents the percentage of global CPU usage.

From Table 3, for the Web1 dataset, we can see that NBCB-e and CBNB-w demonstrate the highest performance, followed by VarLiNGAM. Next are Dynotears and PCGCE. CBNB-e exhibits the lowest score. For the Web2 dataset, NBCB-e attains the highest F1 score, with CBNB-e closely following. The subsequent competitive results are observed with PCGCE and PCMCI$^+$. This could imply that the data is noisy and the time lags are inconsistent throughout time, making the inference of the extended summary graph more robust.

**Antivirus.** Lastly, we consider a dataset which depicts the impacts of antivirus activity in servers which is again provided by EasyVista. This dataset contains 13 time series such that 3 of them are collected with a one-minute sampling rate and the rest with a five-minute sampling rate. The raw data of this case study were initially misaligned. To align them, we use the two pre-processing strategies described in Appendix C, leading to the dataset Antivirus 1 for Strategy 1 and Antivirus 2 for Strategy 2. The two processed datasets consist of 1321 timestamps. The corresponding summary causal graph for Antivirus dataset is presented in Figure 8b where *CUV* represents the percentage of CPU usage of antivirus processes in server V; *CUGV* represents the percentage of CPU usage of the global server V; *MUV* represents the percentage of memory usage of antivirus process; *MUGV* represents the percentage of global memory usage of the server; *RV* represents the Disk IO read in Kbytes/second; *ChIE* refers to the required duration in seconds to open an *IE browser* on server V; *CUP* represents the percentage of CPU usage of antivirus processes in server P; *CUGP* represents the percentage of CPU usage of the global server P; *MUP* represents the percentage of memory usage of antivirus process; *MUGP* represents the percentage of global memory usage of the server; *RP* represents the Disk IO read in Kbytes/second; *ChP* represents refers to the required duration in seconds

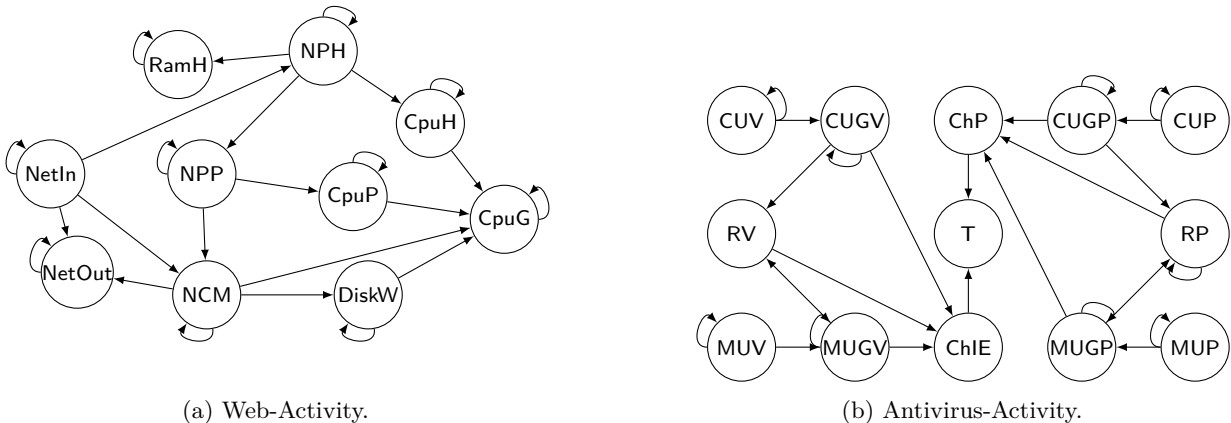

(a) Web-Activity.

(b) Antivirus-Activity.

Figure 8: Summary causal graphs for (a) the Web activity datasets and (b) the Antivirus activity datasets. Those summary causal graphs are constructed by an IT monitoring system's experts.

Table 4: Results for real datasets of Section 5.4 using nonlinear methods. We report the mean and the variance (when meaningful, see data description) of the F1 score of the orientations in the SCG. The best results are in blue bold and the second best results are in green bold.

|  | Temp. | Veil1 | Veil2 | Dairy | Ingest. |
|---|---|---|---|---|---|
| NBCB-w-nl | **1** | **1** | 0 | **0.4** | **0.52** $\pm$ 0.02 |
| CBNB-w-nl | **1** | **1** | 0 | **0.4** | **0.54** $\pm$ 0.06 |
| NBCB-e-nl | **1** | **1** | 1 | **0.4** | 0.43 $\pm$ 0.04 |
| CBNB-e-nl | **1** | **1** | 1 | **0.4** | 0.49 $\pm$ 0.07 |
| PCMCI$^+$-nl | **1** | **1** | 0 | 0.0 | 0.38 $\pm$ 0.09 |
| PCGCE-nl | **0.67** | **1** | 1 | **0.4** | 0.43 $\pm$ 0.03 |

to open a *CITRIX Portal* on server P; $T$ represents the global time in seconds required to open a CITRIX portal and open the IE browser.

From Table 3 for Antivirus 1, we can see that CBNB-e achieves the best result followed by PCGCE and NBCB-e and then by CBNB-w, VarLiNGAM and Dynotears. GCMVL and PCMCI$^+$ have low performance, with PCMCI$^+$ being the worst. For Antivirus 2, we can see that NBCB-e performs best followed by PCGCE and then by CBNB-e. The remaining methods have a substantial drop in performance. As in the case of Web datasets, improved performance of the PCGCE over PCMCI$^+$ suggests that inference of the extended summary graph is more robust in this case. We can also note that NBCB-e and NBCB-w perform well, while VarLiNGAM does not, which could suggest that VarLiNGAM infers a more dense graph.

### 5.4.2 The nonlinear case

The results of the nonlinear counterparts of our hybrid methods and of PCMCI$^+$ and PCGCE on real data are presented in Table 4. In this scenario, we excluded the Web and Antivirus datasets due to computational constraints (running the nonlinear counterparts of the algorithms on these datasets is prohibitively expensive due to their size).

In the temperature dataset, the Veilleux datasets and the Dairy dataset, all considered methods produced consistent results compared to their linear counterparts, except for PCGCE, which exhibited a decrease in performance in the temperature dataset. In the Ingestion mini datasets, we can clearly see that all methods that infer a WCG has a slight increase in performance and all methods that infer an ECG has a slight decrease in performance. In terms of ranking, in the nonlinear case, NBCB-w-nl demonstrates the best performance, followed by CBNB-w-nl, while PCMCI$^+$ performs the worst despite its performance increase.

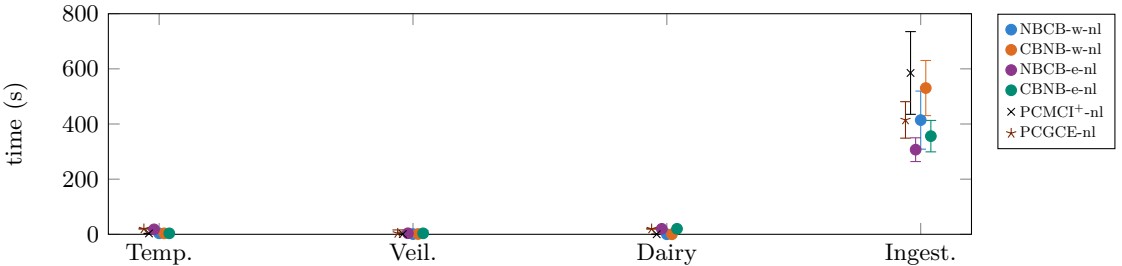

Figure 9: Time computation (in second) for NBCB-w, CBNB-w, NBCB-e, CBNB-e, PCMCI$^+$, and PCGCE for real datasets of Section 5.4. We report the mean and the standard deviation.

The performance improvement observed in methods inferring a WCG can suggest that the dataset Ingestion mini contains nonlinear causal relations. Conversely, the decline seen in methods inferring an ECG could be attributed to the combination of the complexity of nonlinear tests and the necessity for these methods to conduct conditional independence tests with larger conditional sets compared to those considered by other methods.

We also conduct a computation time analysis of the nonlinear counterparts of the methods, as shown in Figure 9. Notably, the computation times for all methods are comparable across the temperature dataset, the Veilleux datasets, and the Dairy dataset, which are relatively small in size. However, in the Ingestion mini datasets, variations in computation time are evident: PCMCI$^+$ exhibits the longest computation time, followed by CBNB-w then by PCGCE and NBCB-w. Our hybrid methods for inferring an ECG (NBCB-e and CBNB-e) exhibit the shortest computation time, with NBCB-e showing a lower computation time compared to CBNB-e. The NBCB methods achieve better time computation because the nonlinear conditional tests they employ are significantly more computationally expensive than learning Gaussian process regression. This implies that the constraint-based step is more costly than the noise-based step. Hence, providing additional knowledge to the constraint-based step, in the form of a causal order, results in a much greater reduction in computation time compared to when the additional knowledge is given to the noise-based step, in the form of the skeleton.

## 6 Discussion

Experiments on simulated data, realistic ecological data, and real data from various applications, show that our hybrid approaches are robust and yield overall good results over all datasets. Notice that for all results on real data, NBCB-w and CBNB-w (which are based on PCMCI$^+$ and VarLiNGAM) never perform simultaneously worse than PCMCI$^+$ and VarLiNGAM. Similarly, NBCB-e and CBNB-e (which are based on PCGCE and VarLiNGAM) never perform simultaneously worse than PCGCE and VarLiNGAM, except for the Web 1 dataset where CBNB-e has the lowest F1-score (but NBCB-e has the best F1-score). In general, NBCB-e and CBCB-e seem to be more reliable than NBCB-w and CBCB-w for the real data we considered, especially when assuming linearity. As mentioned before, the possible explanation is noisy data and inconsistent time lags. In summary, results on simulated data, realistic ecological data, and real data are coherent with the theoretical findings, showing that algorithms from CBNB and NBCB classes are trade-offs between the original methods, potentially exhibiting enhanced performance compared to the original methods when certain assumptions are violated.

In Table 5, we provide various theoretical and experimental criteria to distinguish between the algorithms within the NBCB and CBNB classes. In the second and third rows of Table 5, we present which algorithms infer a WCG and which ones infer an ECG. In the forth and fifth rows, we detail the steps of the NBCB and CBNB classes of methods. The sixth row indicates that NBCB-w and NBCB-e can still be applied even when Assumption 3 is violated. In such cases, the true graph may not be fully retrieved, but it is guaranteed that if the algorithms infer $X \rightarrow Y$, then in the true graph, we can be certain that $Y$ does not cause $X$ (see Proposition 1). Similarly, the seventh row shows that CBNB-w and CBNB-e can be utilized even when

| Class | NBCB | | CBNB | |
|---|---|---|---|---|
| Version | NBCB-e | NBCB-w | CBNB-e | CBNB-w |
| Output | ECG | WCG | ECG | WCG |
| Step 1 | NB1 (VLiNGAM) | | CB1 (PCGCE - PCMCI$^+$) | |
| Step 2 | CB1′ (PCGCE - PCMCI$^+$) | | NB1′ (VLiNGAM) | |
| Violation of Assump. 3 | $(X_t \to Y_t \in \hat{\mathcal{G}}^\star \implies (Y_t \not\to X_t \in \mathcal{G}^\star)$ | | ✗ | ✗ |
| Violation of Assump. 4 | ✗ | ✗ | $(X_{t^\star} - Y_t \in \hat{\mathcal{G}}^\star) \implies (X_{t^\star} - Y_t \in \mathcal{G}^\star)$ | |
| Simulated data (Sec. 5.2) | ✗ | ✓ | ✗ | ✓ |
| Realistic sim. data (Sec. 5.3) | ✓ | ✓ | ✓ | ✓ |
| Real data (Sec. 5.4) | ✓ | ✗ | ✓ | ✗ |

Table 5: Summary of the two proposed classes of methods, NBCB and CBNB. Rows respectively indicate: their versions NBCB-e, NBCB-w, CBNB-e, CBNB-w; outputs; the algorithms used in each step; their theoretical guarantees, or lack thereof (✗), under violation of some assumption; and their advantageous (✓) or limited (✗) performances on the different data scenarios considered in Section 5. $\hat{\mathcal{G}}^\star$ represents the inferred WCG or ECG and $\mathcal{G}^\star$ representes the true WCG or ECG.

Assumption 4 is violated. In this scenario, algorithms are capable of inferring accurate skeleton of the graph, but not the orientations. The last three rows highlight the scenarios where each algorithm outperformed others. Interestingly, in the experimental section, we observed that methods inferring a WCG perform better with simulated data, while methods inferring an ECG excel with real data, assuming linearity.

One of the key limitations of the CBNB and NBCB classes for real-world applications is their reliance on the restrictive assumption that there are no hidden confounders (Assumption 1), which could be often violated. For CBNB, we could consider an extension of the FRITL algorithm (Chen et al., 2021) for time-series, which is based on building the skeleton using FCI extension and refining after using noise-based methods. However, it is not clear how to adapt the necessary conditions for undirected cycle groups. For NBCB it could be more complicated to relax the no hidden confounders assumption.

Additionally, it would be interesting to adapt the CBNB and NBCB classes to the cases when there is a violation of consistency over time or stationarity. One potential direction could be to integrate our methods in the strategy proposed by Saggioro et al. (2020) which combines a causal discovery with a regime learning optimisation approach. However, this direction might require additional assumptions and while it appears promising for CBNB, it is less obvious for NBCB. Another direction could be to assume the presence of an observed contextual variable that explains the non-stationarity (Mooij et al., 2020, Günther et al., 2023).

Finally, adapting these methods for mixed data can be important in many applications. This can be straightforward for the constraint-based part of our algorithms, given the availability of conditional independence tests for mixed data (Zan et al., 2022). However, the adaptation is more challenging for the noise-based part.

# 7 Conclusion

In this paper, we introduced a framework for hybrids of noise-based and constraint-based methods that can discover causal graphs from temporal data. Algorithms in the first class, denoted NBCB, start with ordering instantaneous relations, and then prune edges of the fully oriented graph. On the other hand, the algorithms from the second class, denoted CBNB, start by finding the skeleton and the orientation of lagged relations using a constraint-based method and temporal priority, then orient instantaneous relations by ordering nodes in each cycle group of instantaneous relations. Overall, the performance of our algorithms is a trade-off between the performance of constraint-based and noise-based algorithms when all assumptions are satisfied, and they outperform other methods in the cases where some of the assumptions are violated.

For future works, it would be interesting to extend these approaches to cases involving hidden confounders, non-stationarity, and mixed data.

**Acknowledgments**

We thank Ali Aït-Bachir, Christophe de Bignicourt and Rachid Mokhtari from EasyVista for providing the IT monitoring data along with the underlying causal graphs. We also thank Giovanni Poggiato for several discussions about the realistic ecological simulated data. Finally, we thank the anonymous reviewers for their helpful comments. This work was partially supported by the CIPHOD project (ANR-23-CPJ1-0212-01), by MIAI@Grenoble Alpes (ANR-19-P3IA-0003), by the Horizon Europe Obsgession project (No: 101134954) and the FRB-CESAB through the IMPACT working group.

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

# A  Proofs

We recall the definition of blocked paths, backdoor paths, and the backdoor criterion.

**Definition 7** (Blocked path, Pearl, 2000). *A path is said to be* blocked *by a set of nodes* $\mathbb{S} \subset \mathbb{V}$ *if it contains an intermediate cause or a common cause $X$ such that $X \in \mathbb{S}$ and if it contains a collider $X$ such that $X \notin \mathbb{S}$ and no descendant of $X$ is in $\mathbb{S}$.*

Note that a path that is not blocked is said to be active.

**Definition 8** (Backdoor path, Pearl, 2000). *A path between an ordered pair $(X, Y)$ is said to be a* backdoor path *between $X$ and $Y$ if it contains an arrow into $X$.*

Note that blocking all backdoor paths between two nodes eliminates confounding bias (Pearl, 2000).

**Theorem 1.** *Let $\mathcal{G}^f = (\mathbb{V}^f, \mathbb{E}^f)$ be an FTCG. Under Assumptions 1, 2, 3, 4, 5 and given perfect conditional independence information about all pairs of variables in $\mathbb{V}^f$, any algorithm in the NBCB class returns the correct WCG or the correct ECG compatible with $\mathcal{G}^f$.*

*Proof.* This proof is similar to the proof of Theorem 4 in Assaad et al. (2021) for NBCB$^{\mathrm{acyclic}}$. Given the Assumption 4 and causal sufficiency and assuming NB1 is consistent with Assumption 4, NB1 would infer the correct causal order $\hat{\pi}$. Given the causal order $\hat{\pi}$ and temporal priority, we can orient all edges in a fully connected graph, which represents a super graph that contains the true graph. Given Assumptions 1, 2, 3, 5 and assuming that the constraint-based algorithm on which CB1$'$ is based on is sound and complete, if we do not consider the causal order, CB1 would prune all unnecessary edges by removing edges between two nodes that are conditionally independent given a subset $\mathbb{S}$ adjacent to one of these two nodes and yield the correct skeleton. Given the causal order $\hat{\pi}$, the subset $\mathbb{S}$ can be reduced by containing only parents (instead of adjacencies). Thus, again by Assumptions 1, 2, 3, removing all edges between the conditionally independent nodes, the only edges that will be left with are causal, and so the graph would be correct. $\square$

**Proposition 1** (Violation of Assumption 3). *Under Assumptions 1, 2, 5, given that the SCM is a restrictive additive noise model (Peters et al., 2014), and given a correct causal order between instantaneous nodes, the NBCB class would give a WCG or an ECG such that, for each pair of nodes $X_{t^\star}$ and $Y_t$, one of the following possibilities holds true:*

*(1)  The causal relationship between $X_{t^\star}$ and $Y_t$ is correctly identified.*

*(2)  $X_{t^\star}$ and $Y_t$ are not adjacent in the inferred graph, but they are adjacent in the true graph.*

*(3)  $X_{t^\star}$ and $Y_t$ are adjacent in the inferred graph, but they are not adjacent in the true graph.*

*Proof.* Given that the noise-based algorithm on which the NB1 step is based on is correct, the NB1 step would give the correct causal order. Thus, having correct causal order for the instantaneous nodes and using orientation by time, the NB1 step would give the fully connected oriented graph, such that each edge that is present in the true graph is correctly oriented. The second step involves the CB1$'$ step for pruning the edges. Since we consider that adjacency faithfulness is violated, we can have one of the following cases:

- If a pair of nodes $X_{t^\star}$ and $Y_t$ is adjacent in the true graph there are two possible cases:
  - (a) there exist set $\mathbb{S}$ such that $X_{t^\star} \perp\!\!\!\perp Y_t \mid \mathbb{S}$. In this case, CB1$'$ would erroneously remove the edge between $X_{t^\star}$ and $Y_t$ in the inferred graph and we obtain case (2) in the proposition
  - (b) there exists no set $\mathbb{S}$ such that $X_{t^\star} \perp\!\!\!\perp Y_t \mid \mathbb{S}$, in this case, the CB1$'$ would keep this edge and orientation of this edge is correct (given step NB1), this edge corresponds to correctly inferred causal relationship case (1)

- If a pair of nodes $X_{t^\star}$ and $Y_t$ is not adjacent in the true graph but they are connected by an active path $u = \langle X_{t^\star}, V_{t_2^\star}^2, \cdots, V_{t_{n-1}^\star}^{n-1}, Y_t \rangle$ of size $n > 2$. Suppose that due to violation of adjacency faithfulness, the step CB1$'$ removes the edge $X_{t^\star} - V_{t_2^\star}^2$ and the edge $V_{t_{n-1}^\star}^{n-1} - Y_t$. If $n = 3$, then

CB1′ will never test if $X_{t^\star} \perp\!\!\!\perp Y_t \mid \mathbb{S}$, such that $V^2_{t_2^\star} \in \mathbb{S}$ therefore it will not remove the edge between $X_{t^\star}$ and $Y_t$ in the inferred graph. If $n > 3$ and if CB1′ removes all possible edges between $X_{t^\star}$ and each node in $\{V^3_{t_3^\star}, \cdots, V^{n-1}_{t_{n-1}^\star}\}$ and all possible edges between $Y_t$ and each node in $\{V^2_{t_2^\star}, \cdots, V^{n-2}_{t_{n-2}^\star}\}$ in the inferred graph (due to a some specific configuration of the parameters that violates faithfulness but not adjacency faithfulness). Then, in this case, CB1′ will never test if $X_{t^\star} \perp\!\!\!\perp Y_t \mid \mathbb{S}$, such that $\{V^2_{t_2^\star}, \cdots, V^{n-1}_{t_{n-1}^\star}\} \cap \mathbb{S} = \emptyset$ therefore it will not remove the edge between $X_{t^\star}$ and $Y_t$ in the inferred graph.

$\square$

**Theorem 2.** *Let $\mathcal{G}^f = (\mathbb{V}^f, \mathbb{E}^f)$ be an FTCG. Under Assumptions 1, 2, 3, 4, 5 and given perfect conditional independence information about all pairs of variables in $\mathbb{V}^f$, any algorithm in the CBNB class returns the correct WCG or the correct ECG compatible with $\mathcal{G}^f$.*

*Proof.* Without any given causal order, CB1 uses the first two steps of the constraint-based algorithm (omitting the orientation step). Using the full constraint-based algorithm under Assumptions 1, 2 and 3, we obtain correct partially complete partially oriented WCG or ECG $\hat{\mathcal{G}}^\star$, i.e., correct skeleton, all instantaneous relations are not oriented and all lagged relations are oriented. So, having $\hat{\mathcal{G}}^\star$, what is left to prove is that applying NB1′ on the nodes $\mathbb{I}_t$ that belong to each of undirected cycle group $\mathbb{C}$ given the past parents $Pa_{\hat{\mathcal{G}}^\star}(\mathbb{I}_t) \backslash \mathbb{V}_t$ is free of confounding bias.

To obtain the correct causal order $\hat{\pi}$ between nodes $\mathbb{I}_t$, we need to verify that in the subgraph of $\hat{\mathcal{G}}^\star$ containing the nodes $\mathbb{I}_t \cup Pa_{\hat{\mathcal{G}}^\star}(\mathbb{I}_t)$ (which is by definition acyclic) for every two adjacent nodes $X_t, Y_t \in \mathbb{I}_t$, there exists a set $\mathbb{S} \subseteq \mathbb{I}_t \backslash \{X_t, Y_t\} \cup Pa_{\hat{\mathcal{G}}^\star}(\mathbb{I}_t)$, such that $\mathbb{S}$ blocks all backdoor paths between $X_t$ and $Y_t$ (if there is no edge between nodes $X_t$ and $Y_t$, then there is no orientation to be determined by NB1′). Suppose $X_t \to Y_t$ in the true graph (but in the output of CB1 this edge is unoriented) and there exists some active backdoor path $u$ between $X_t$ and $Y_t$, we consider two cases:

(a) Suppose all nodes in path $u$ belong to $\mathbb{V}_t$. By definition of an undirected cycle group $\mathbb{C}$, all nodes in $u$ are also in $\mathbb{I}_t$, which means that the common cause (common ancestor) on the path is also in $\mathbb{I}_t$, i.e., causal sufficiency is satisfied. This means, that $u$ can be blocked by a node in $\mathbb{I}_t$. Thus there exists $\mathbb{S} \subseteq \mathbb{I}_t \backslash \{X_t, Y_t\}$ such that all backdoor paths between $X_t$ and $Y_t$ are blocked.

(b) Suppose that some nodes in $u$ belong to $\mathbb{V}^\star \backslash \mathbb{V}_t$. In this case, conditioning on $Pa_{\hat{\mathcal{G}}^\star}(\mathbb{I}_t)$ blocks $u$ since $Pa_{\hat{\mathcal{G}}^\star}(\mathbb{I}_t)$ are the parents of $\mathbb{I}_t$ and none of the nodes in $Pa_{\hat{\mathcal{G}}^\star}(\mathbb{I}_t)$ are colliders of any two nodes in $\mathbb{I}_t$. Thus all backdoor paths between $X_t$ and $Y_t$ passing by $\mathbb{V}^\star \backslash \mathbb{V}_t$ can be blocked by $Pa_{\hat{\mathcal{G}}^\star}(\mathbb{I}_t)$ and all the backdoor paths that are left are the ones discussed in (a).

$\square$

**Proposition 2** (Violation of Assumption 4). *Under Assumptions 1, 2, 3, 5 and given perfect conditional independence information about all pairs of variables, CBNB is guaranteed to find the correct skeleton of the WCG or the ECG.*

*Proof.* Given that the CB algorithm used for the CBNB method is correct under Assumptions 1 and 2, the result of the first step CB1 would give the correct skeleton under the Assumption 3. $\square$

For example, if the PCMCI$^+$ method is used for the CBNB method, then the correctness of the skeleton comes from Theorem 1 in Runge (2020). In case when CB1 is based on PCGCE algorithm, the correctness of the skeleton is shown in Theorem 1 in Assaad et al. (2022c).

# B  Pseudo-code algorithms

In our experimental section, we used an NB1 and an NB1′ steps based on the VarLiNGAM algorithm (NB1 in NBCB-w and NBCB-e and NB1′ in CBNB-w and CBNB-e) and we used respectively a CB1 and a CB1′ steps based on the PCMCI$^+$ algorithm (CB1 in CBNB-w and CB1′ in NBCB-w) and on the PCGCE algorithm (CB1 in CBNB-e and CB1′ in NBCB-e). Since NB1′ and CB′ steps require more modifications compared to NB1 and CB1 steps starting from the initial methods, in the following, we provide the pseudo-codes of each NB′ and CB′ steps that we used. But first, we start by briefly recalling VarLiNGAM, PCMCI$^+$, and PCGCE algorithms which all assume causal sufficiency (Assumption 1) while pointing out either their NB1 or CB1 step.

VarLiNGAM (Hyvärinen et al., 2008) is a noise-based causal discovery algorithm for time series data that constructs a WCG. First, it estimates a classic autoregressive model for the data using any conventional implementation of a least-squares method. It then computes the residuals and then performs the LiNGAM analysis (Shimizu et al., 2006; 2011) on the residuals. Note that the LiNGAM analysis can be either done using the ICALiNGAM (Shimizu et al., 2006) or DirectLiNGAM (Shimizu et al., 2011), in this work, we use DirectLiNGAM. This step, which we refer to as NB1, gives the causal order and the estimate of the instantaneous causal effects. After that, it computes the estimates of lagged causal effects. Finally, it estimates redundant directed edges to find the underlying WCG.

PCMCI$^+$ (Runge, 2020) is a constraint-based causal discovery algorithm for time series data that constructs a WCG. First, the PC1 lagged phase infers a superset of the lagged parents together with the parents of instantaneous ancestors. Next, the MCI instantaneous phase starts with links found in the previous step and all possible instantaneous links, then it conducts momentary conditional independence (MCI) with a modified conditioning set learned in the previous step to increase detection power. This step, which we refer to as CB1, gives a partially oriented graph where lagged relations are oriented and where instantaneous are non-oriented. Finally, it orients edges using the same rules used in the PC-algorithm (Spirtes et al., 2000, Meek, 1995).

Similarly, PCGCE (Assaad et al., 2022c) is also a constraint-based causal discovery algorithm for time series data, but that constructs an ECG without passing by a WCG. It also consists of two steps. First, it searches for the skeleton of the ECG using a procedure similar to the PC-algorithm that is order-independent by using a conditional independence test between either two nodes in the present slice or one node in the present slice and one node in the past slice, which can be multidimensional. Similarly to the case of PCMCI$^+$, this step, which we refer to as CB1, gives a partially oriented graph where lagged relations are oriented and where instantaneous are non-oriented. Then it orients edges using the same rules used in the PC-algorithm (Spirtes et al., 2000, Meek, 1995).

In the following, we present the pseudo-codes of NB1′ based on VarLiNGAM, CB1′ based on PCMCI$^+$ and CB1′ based on PCGCE. We colour in orange the parts that are different from the initial algorithms. Remark that the orange colour indicates that the corresponding parts are added or modified compared to the initial algorithms, but they do not indicate parts of the initial algorithms that were deleted.

## B.1  NB1′ based on VarLiNGAM (Algorithm 3)

The NB1′ step based on VarLiNGAM is almost identical to the NB1 step based on VarLiNGAM. As NB1, NB1′ starts by computing the residuals of all instantaneous nodes by regressing them on their past. However, unlike NB1, NB1′ focuses only on a subset of instantaneous nodes $\mathbb{I}_t \subseteq \mathbb{V}_t$. Note that also unlike NB1, NB1′ takes as input a partially oriented graph $\hat{\mathcal{G}}^\star$ (the output of CB1 step which has the correct skeleton) and that $\mathbb{I}_t \cup Pa_{\hat{\mathcal{G}}^\star}(\mathbb{I}_t)$ should satisfy causal sufficiency. By construction, causal sufficiency is satisfied when $\mathbb{I}_t$ is an undirected cycle group as defined in Definition 6. The pseudo-code of NB1′ is provided in Algorithm 3.

## B.2  CB1′ based on PCMCI$^+$ (Algorithm 4)

CB1 and CB1′ based on the PCMCI$^+$ algorithm use conditional independence test CI() that returns at the same time the p-value and the statistic of the test. The main differences between the CB1′ and the CB1 step

---

**Algorithm 3:** NB1′ based on VarLiNGAM (parts in orange are different from the initial algorithms)

---

**Input:** A multivariate time series, a maximal temporal lag $\gamma$, a significance threshold $\alpha$, an independence measure I(), the output of the CB1 step $\hat{\mathcal{G}}^{\star}$ (partially oriented), and instantaneous nodes of interest $\mathbb{I}_t \subseteq \mathbb{V}_t$

**Result:** $\hat{\mathcal{G}}^{\star}$ (fully oriented)

**if** $\hat{\mathcal{G}}^{\star}$ *is an ECG* **then**

   Construct a WCG $\mathcal{G}^{\mathrm{w}} = (\mathbb{E}^{\mathrm{w}}, \mathbb{V}^{\mathrm{w}} = \{\mathbb{V}_{t-\gamma}, \cdots, \mathbb{V}_t\})$ s.t. $\forall X_{t-} \in \mathbb{V}_{t-}, Y_t \in \mathbb{V}_t$, if $X_{t-} \to Y_t \in \mathbb{E}^{\mathrm{e}}$ then $\forall \ell \in \{1, \cdots, \gamma\}$, $X_{t-\ell} \to Y_t \in \mathbb{E}^{\mathrm{w}}$ and $\forall X_t, Y_t \in \mathbb{V}_t$ if $X_t \neq Y_t$, $X_t \to Y_t \in \mathbb{E}^{\mathrm{e}}$ then $X_t \neq Y_t$, $X_t \to Y_t \in \mathbb{E}^{\mathrm{w}}$;

**else**

   $\hat{\mathcal{G}}^w = \hat{\mathcal{G}}^{\star}$;

**for** $Y_t \in \mathbb{I}_t$ **do**

   Estimate a classic autoregressive model for the data

$$Y_t = \sum_{X_{t-\ell} \in Pa_{\hat{\mathcal{G}}^w}(\mathbb{I}_t)} a_{xy\ell} X_{t-\ell} + \xi_t^y$$

   using any conventional implementation of a least-squares method. Note that here $\ell > 0$, so it is really a classic AR model;
   Compute the residuals, that is, estimates of $\xi_t^y$ ;

$$\hat{\xi}_t^y = Y_t - \sum_{X_{t-\ell} \in Pa_{\hat{\mathcal{G}}^w}(\mathbb{I}_t)} \hat{a}_{xy\ell} X_{t-\ell}$$

Initialize a bijective mapping function $\pi$;
$i = 1$;
Initialize a list $\mathbb{S}$ containing all nodes in $\mathbb{I}_t$;
**while** size($\mathbb{S}$) $> 1$ **do**

   Initialize an empty list $\mathbb{H}$;
   **for** $X_t \in \mathbb{S}$ **do**

      **for** $Y_t \in \mathbb{S}\backslash\{X_t\}$ **do**

         Perform least squares regressions of $\hat{\xi}_t^x$ on $\hat{\xi}_t^y$ and compute the residuals:

$$\hat{\epsilon}^{Y_t} = \hat{\xi}_t^y - \frac{\mathrm{cov}(\hat{\xi}_t^x, \hat{\xi}_t^y)}{\mathrm{var}(\hat{\xi}_t^x)}$$

      Estimate the dependence between the total residuals and $X_t$:

$$h = \sum_{Y_t \in \mathbb{S}\backslash\{X_t\}} \mathrm{I}(\hat{\xi}_t^x, \hat{\epsilon}^{Y_t})$$

      Append $h$ to the end of $\mathbb{H}$;

   Find the node $X_t$ corresponding to $\hat{\xi}_t^x$ that is most independent of its residuals in $\mathbb{H}$;
   $\pi(X_t) = i$;
   $i = i + 1$;
   Remove $X_t$ from $\mathbb{S}$;

$\pi(X_t) = i$ where $X_t$ is the remaining instantaneous node in $\mathbb{S}$;
Orient $\hat{\mathcal{G}}^{\star}$ s.t. $\forall X_t - Y_t \in \mathbb{E}^{\star}$, $X_t \to Y_t \in \mathbb{E}^{\star}$ if $\pi(X_t) < \pi(Y_t)$;

---

based on PCMCI$^+$ is that CB1$'$ takes a causal order as input, and therefore, it starts with a fully-oriented graph, in addition in the MCI instantaneous phase, it conditions only using parents (PCMCI$^+$ condition also on instantaneous adjacencies).

---

**Algorithm 4:** CB1$'$ based on PCMCI$^+$ (parts in orange are different from the initial algorithms)

---

**Input:** A multivariate time series, a maximal temporal lag $\gamma$ and a significance threshold $\alpha$, a conditional independence test CI(), and a causal order $\pi$

**Result:** $\mathcal{G}^{\mathrm{w}}$ (WCG)

Construct an fully-connected WCG $\mathcal{G}^{\mathrm{w}} = (\mathbb{E}^{\mathrm{w}}, \mathbb{V}^{\mathrm{w}} = \{\mathbb{V}_{t-\gamma}, \cdots, \mathbb{V}_t\})$ s.t.

$\forall X_{t-\ell} \in \{\mathbb{V}_{t-\gamma}, \cdots, \mathbb{V}_{t-1}\}, Y_t \in \mathbb{V}_t, X_{t-\ell} \to Y_t \in \mathbb{E}^{\mathrm{w}}$ and $\forall X_t, Y_t \in \mathbb{V}_t$ s.t. $X_t \neq Y_t$, $X_t \to Y_t \in \mathbb{E}^{\mathrm{w}}$ if $\pi(X_t) < \pi(Y_t)$;

**for** $Y_t \in \mathbb{V}_t$ **do**

    Initialize $\hat{B}_t(Y_t) = \mathbb{V}^{\mathrm{w}} \backslash \mathbb{V}_t$;

    Initialize $I^{\min}(X_{t-\ell}, Y_y) = \infty$ $\forall X_{t-\ell} \in \hat{B}_t(Y_t)$;

    $n = 0$;

    **while** $\exists X_{t-\ell} \in \hat{B}_t(Y_t)$ *s.t.* size$(\hat{B}_t(Y_t)) \geq n$ **do**

        **for** $X_{t-\ell} \in \mathbb{V}^{\mathrm{w}} \backslash \mathbb{V}_t$ *s.t.* size$(\hat{B}_t(Y_t)) \geq n$ **do**

            $\mathbb{S} =$ first $n$ nodes in $\hat{B}_t(Y_t)$;

            $p, h = CI(X_{t-\ell}, Y_t \mid \mathbb{S})$;

            $I^{\min}(X_{t-\ell}, Y_t) = \min(|h|, I^{\min}(X_{t-\ell}, Y_t))$;

            **if** $p > \alpha$ **then**

                mark $X_{t-\ell}$ for removal

        $\forall X_{t-\ell}$ marked for removal, remove $X_{t-\ell} \to Y_t$ from $\mathbb{E}^{\mathrm{w}}$;

        Sort $\hat{B}_t(Y_t)$ by $I^{\min}(X_{t-\ell}, Y_y)$ from largest to smallest;

        $n = n + 1$;

Initialize $I^{\min}(X_{t-\ell}, Y_y) = \infty$ $\forall X_{t-\ell} \in \hat{B}_t(Y_t)$;

$n = 0$;

**while** $\exists X_{t-\ell} - Y_t \in \mathbb{E}^{\mathrm{w}}$ $\forall \ell \geq 0$ *s.t.* size$(Pa_{\hat{\mathcal{G}}^w}(Y_t) \cap \mathbb{V}_t \backslash \{X_{t-\ell}\}) \geq n$ **do**

    **for** $X_{t-\ell} - Y_t \in \mathbb{E}^{\mathrm{w}}$ $\forall \ell \geq 0$ *s.t.* size$(Pa_{\hat{\mathcal{G}}^w}(Y_t) \cap \mathbb{V}_t \backslash \{X_{t-\ell}\}) \geq n$ **do**

        **while** $\exists X_{t-\ell} - Y_t \in \mathbb{E}^{\mathrm{w}}$ *and not all* $\mathbb{S} \in Pa_{\hat{\mathcal{G}}^w}(Y_t) \cap \mathbb{V}_t \backslash \{X_{t-\ell}\}$ *with* size$(\mathbb{S}) = n$ *have been considered* **do**

            **for** $\mathbb{S} \in Pa_{\hat{\mathcal{G}}^w}(Y_t) \cap \mathbb{V}_t \backslash \{X_{t-\ell}\}$ *s.t.* size$(\mathbb{S}) = n$ **do**

                $p, h = CI(Y_t, X_{t-\ell} \mid \mathbb{S}, \hat{B}_t(Y_t) \backslash \{X_{t-\ell}\}, \hat{B}_{t-\ell}(X_{t-\ell}))$;

                $I^{\min}(X_{t-\ell}, Y_t) = \min(|h|, I^{\min}(X_{t-\ell}, Y_t))$;

                **if** $p > \alpha$ **then**

                    Remove $X_{t-\ell} \to Y_t$ from $\mathbb{E}^{\mathrm{w}}$ or $X_{t-\ell} - Y_t$ from $\mathbb{E}^{\mathrm{w}}$;

    $n = n + 1$;

    Sort $Pa_{\hat{\mathcal{G}}^w}(Y_t) \cap \mathbb{V}_t$ by $I^{\min}(X_{t-\ell}, Y_y)$ from largest to smallest;

---

### B.3 CB1$'$ based on PCGCE (Algorithm 5)

CB1 and CB1$'$ based on the PCGCE algorithm use conditional independence test CI() that returns either the p-value of the test or the statistic without computing the p-value. The main difference between the CB1$'$ and the CB1 step based on the PCGCE algorithim is that CB1$'$ takes a causal order as input, and therefore, it starts with a fully-oriented graph and in addition, it conditions only using parents (PCGCE condition on adjacencies).

---

**Algorithm 5:** CB1′ based on PCGCE (parts in orange are different from the initial algorithms)

---

**Input:** A multivariate time series, a maximal temporal lag $\gamma$ and a significance threshold $\alpha$, a
  conditional independence test CI(), and a causal order $\pi$

**Result:** $\mathcal{G}^e$ (ECG)

Construct an fully-connected ECG $\mathcal{G}^e = (\mathbb{E}^e, \mathbb{V}^e = \{\mathbb{V}_{t-}, \mathbb{V}_t\})$ s.t. $\forall X_{t-} \in \mathbb{V}_{t-}, Y_t \in \mathbb{V}_t, X_{t-} \rightarrow Y_t \in \mathbb{E}^e$

 and $\forall X_t, Y_t \in \mathbb{V}_t$ s.t. $X_t \neq Y_t$, $X_t \rightarrow Y_t \in \mathbb{E}^e$ if $\pi(X_t) < \pi(Y_t)$ ;

$n = 0$;

**while** $\exists X_{t*} - Y_t \in \mathbb{E}^e \ \forall t* \in \{t, t-\}$ *s.t.* $\text{size}(Pa_{\hat{\mathcal{G}}^e}(Y_t) \backslash \{X_{t*}\}) \geq n$ **do**

    Initialize $\mathbb{D}$ and $\mathbb{H}$ as empty lists;

    **for** $X_{t*} - Y_t \in \mathbb{E}^e \ \forall t* \in \{t, t-\}$ *s.t.* $\text{size}(Pa_{\hat{\mathcal{G}}^e}(Y_t) \backslash \{X_{t*}\}) = n$ **do**

        **while** $\exists X_{t*} - Y_t \in \mathbb{E}^e$ *and not all* $\mathbb{S} \in Pa_{\hat{\mathcal{G}}^e}(Y_t) \backslash \{X_{t*}\}$ *with* $\text{size}(\mathbb{S}) = n$ *have been considered* **do**

            **for** $\mathbb{S} \subset Pa_{\hat{\mathcal{G}}^e}(Y_t) \backslash \{X_{t*}\}$ *s.t.* $\text{size}(\mathbb{S}) = n$ **do**

                $-, h = CI(X_{t*}, Y_t \mid \mathbb{S})$;

                Save $(X_{t*}, Y_t, \mathbb{S})$ in $\mathbb{D}$ and $h$ in $\mathbb{H}$

    Sort $\mathbb{D}$ and $\mathbb{H}$ by $\mathbb{H}$ from smallest to largest;

    **for** $X_{t*}, Y_t, \mathbb{S} \in \mathbb{D}$ *s.t.* $\mathbb{S} \subseteq Pa_{\hat{\mathcal{G}}^e}(Y_t)$ **do**

        $p, - = CI(X_{t*}, Y_t \mid \mathbb{S})$;

        **if** $p > \alpha$ **then**

            Remove $X_{t*} \rightarrow Y_t$ from $\mathbb{E}^e$ or $X_{t*} - Y_t$ from $\mathbb{E}^e$;

    $n = n + 1$;

---

## C  Experimental setup

Time series in monitoring systems are not always exactly aligned together and come in different sampling rates as the timestamps depend on when the data was collected. In the following, we present two pre-processing strategies that we considered for aligning time series:

- Strategy 1: Time series are analyzed in terms of sampling rates and the lowest one is chosen. Afterwards, all the time series are re-sampled according to this lowest sampling rate with the closest value to the timestamp taken as the new value. Upon re-sampling, missing values can be clearly observed. If missing values are detected, they are filled using simple linear interpolation of Pandas data frames[6].

- Strategy 2: Each raw value $x_i$ is converted into integral value $s_i$ at each point $i$ as follows: $s_i = x_i(t_i - t_{i-1}) + s_{i-1}$. Then all time series are re-sampled such that each re-sampled value $x_j$ at every $n$ (the lowest sampling rate) steps is calculated as follows: $x_j = \frac{s_i - s_{i-n}}{t_i - t_{i-n}}$. The time $t_i$ (of value $s_i$) is the time that is after the corresponding time to $x_j$.

## D  Links to datasets

**Temperature.** Available at https://webdav.tuebingen.mpg.de/cause-effect/.

**Veilleux.** Available at http://robjhyndman.com/tsdldata/data/veilleux.dat.

**Diary.** Available at http://future.aae.wisc.edu.

**Ingestion mini.** Available at https://easyvista2015-my.sharepoint.com/personal/aait-bachir_easyvista_com/_layouts/15/onedrive.aspx?id=%2Fpersonal%2Faait%2Dbachir%5Feasyvista%5Fcom%2FDocuments%2FLab%2FPublicData&ga=1.

---

[6]https://pandas.pydata.org/docs/reference/api/pandas.DataFrame.interpolate.html

**Web.** Available at https://easyvista2015-my.sharepoint.com/personal/aait-bachir_ easyvista_com/_layouts/15/onedrive.aspx?id=%2Fpersonal%2Faait%2Dbachir%5Feasyvista%5Fcom% 2FDocuments%2FLab%2FPublicData&ga=1.

**Antivirus.** Available at https://easyvista2015-my.sharepoint.com/personal/aait-bachir_ easyvista_com/_layouts/15/onedrive.aspx?id=%2Fpersonal%2Faait%2Dbachir%5Feasyvista%5Fcom% 2FDocuments%2FLab%2FPublicData&ga=1.

## E   Additional experiments

In Table 6 we provide F1 score on the adjacencies. In contrast to Table 1 in the main text, this Table shows the performance of the algorithms on skeleton recovery which allows to illustrate the robustness of CBNB class to Assumption 4.

Table 6: Results obtained on the simulated data of Section 5.2 for the different structures with 1000 observations with non-Gaussian noise (top panel) and with Gaussian noise (bottom panel). We report the mean and the standard deviation of the F1 score on adjacencies. The best results are in blue bold and the second best results are in green bold.

|  | Diamond | Cyclic Diamond | Unf. Diamond | Adj. Unf. Diamond |
|---|---|---|---|---|
| | Non-Gaussian noise | | | |
| NBCB-w | **0.95** ± 0.01 | 0.95 ± 0.01 | 0.94 ± 0.01 | **0.88** ± 0.01 |
| CBNB-w | **0.95** ± 0.01 | **0.96** ± 0.01 | 0.95 ± 0.01 | **0.88** ± 0.01 |
| NBCB-e | 0.86 ± 0.01 | 0.85 ± 0.01 | **0.97** ± 0.01 | **0.87** ± 0.01 |
| CBNB-e | 0.85 ± 0.01 | 0.86 ± 0.01 | **0.97** ± 0.01 | **0.87** ± 0.01 |
| GCMVL | 0.87 ± 0.01 | 0.92 ± 0.01 | 0.07 ± 0.01 | 0.06 ± 0.02 |
| PCMCI$^+$ | **0.95** ± 0.01 | **0.96** ± 0.01 | 0.95 ± 0.04 | **0.88** ± 0.01 |
| PCGCE | 0.85 ± 0.01 | 0.86 ± 0.01 | **0.97** ± 0.01 | **0.87** ± 0.01 |
| Dynotears | 0.09 ± 0.03 | 0.0 ± 0.0 | 0.0 ± 0.0 | 0.0 ± 0.0 |
| VarLiNGAM | **0.99** ± 0.01 | **0.97** ± 0.01 | **0.98** ± 0.01 | **0.87** ± 0.01 |
| | Gaussian noise | | | |
| NBCB-w | **0.93** ± 0.01 | **0.93** ± 0.01 | 0.90 ± 0.01 | **0.87** ± 0.01 |
| CBNB-w | **0.96** ± 0.01 | **0.96** ± 0.01 | **0.94** ± 0.01 | **0.87** ± 0.01 |
| NBCB-e | 0.83 ± 0.01 | 0.83 ± 0.01 | 0.93 ± 0.01 | **0.85** ± 0.01 |
| CBNB-e | 0.85 ± 0.01 | 0.84 ± 0.01 | **0.97** ± 0.01 | **0.87** ± 0.01 |
| GCMVL | 0.88 ± 0.01 | 0.91 ± 0.01 | 0.02 ± 0.01 | 0.03 ± 0.01 |
| PCMCI$^+$ | **0.96** ± 0.01 | **0.96** ± 0.01 | **0.94** ± 0.01 | **0.87** ± 0.01 |
| PCGCE | 0.85 ± 0.01 | 0.84 ± 0.01 | **0.97** ± 0.01 | **0.87** ± 0.01 |
| Dynotears | 0.13 ± 0.05 | 0.0 ± 0.0 | 0.0 ± 0.0 | 0.0 ± 0.0 |
| VarLiNGAM | 0.91 ± 0.01 | 0.92 ± 0.01 | 0.93 ± 0.01 | 0.84 ± 0.01 |

