** $\color{orange}X_{t*} - Y_t \in \mathbb{E}^{\mathrm{e}}\color{black}$ $\forall t* \in \{t, t-\}$ *s.t.* $\mathrm{size}(\color{orange}Pa_{\hat{\mathcal{G}}^e}(Y_t)\color{black}\backslash\{X_{t*}\}) = n$ **do**

   **while** $\exists X_{t*} - Y_t \in \mathbb{E}^{\mathrm{e}}$ *and not all* $\mathbb{S} \in \color{orange}Pa_{\hat{\mathcal{G}}^e}(Y_t)\color{black}\backslash\{X_{t*}\}$ *with* $\mathrm{size}(\mathbb{S}) = n$ *have been considered* **do**

    **for** $\mathbb{S} \subset \color{orange}Pa_{\hat{\mathcal{G}}^e}(Y_t)\color{black} \backslash \{X_{t*}\}$ *s.t.* $\mathrm{size}(\mathbb{S}) = n$ **do**

     $-, h = CI(X_{t*}, Y_t \mid \mathbb{S})$;

     Save $(X_{t*}, Y_t, \mathbb{S})$ in $\mathbb{D}$ and $h$ in $\mathbb{H}$

  Sort $\mathbb{D}$ and $\mathbb{H}$ by $\mathbb{H}$ from smallest to largest;

  **for** $X_{t*}, Y_t, \mathbb{S} \in \mathbb{D}$ *s.t.* $\mathbb{S} \subseteq \color{orange}Pa_{\hat{\mathcal{G}}^e}(Y_t)\color{black}$ **do**

   $p, - = CI(X_{t*}, Y_t \mid \mathbb{S})$;

   **if** $p > \alpha$ **then**

    Remove $X_{t*} \to Y_t$ from $\mathbb{E}^{\mathrm{e}}$ or $X_{t*} - Y_t$ from $\mathbb{E}^{\mathrm{e}}$;

  $n = n + 1$;

---

## C   Experimental setup

Time series in monitoring systems are not always exactly aligned together and come in different sampling rates as the timestamps depend on when the data was collected. In the following, we present two pre-processing strategies that we considered for aligning time series:

- Strategy 1: Time series are analyzed in terms of sampling rates and the lowest one is chosen. Afterwards, all the time series are re-sampled according to this lowest sampling rate with the closest value to the timestamp taken as the new value. Upon re-sampling, missing values can be clearly observed. If missing values are detected, they are filled using simple linear interpolation of Pandas data frames[6].

- Strategy 2: Each raw value $x_i$ is converted into integral value $s_i$ at each point $i$ as follows: $s_i = x_i(t_i - t_{i-1}) + s_{i-1}$. Then all time series are re-sampled such that each re-sampled value $x_j$ at every $n$ (the lowest sampling rate) steps is calculated as follows: $x_j = \frac{s_i - s_{i-n}}{t_i - t_{i-n}}$. The time $t_i$ (of value $s_i$) is the time that is after the corresponding time to $x_j$.

## D   Links to datasets

**Temperature.** Available at https://webdav.tuebingen.mpg.de/cause-effect/.

**Veilleux.** Available at http://robjhyndman.com/tsdldata/data/veilleux.dat.

**Diary.** Available at http://future.aae.wisc.edu.

**Ingestion mini.** Available at https://easyvista2015-my.sharepoint.com/personal/aait-bachir_easyvista_com/_layouts/15/onedrive.aspx?id=%2Fpersonal%2Faait%2Dbachir%5Feasyvista%5Fcom%2FDocuments%2FLab%2FPublicData&ga=1.

---

[6] https://pandas.pydata.org/docs/reference/api/pandas.DataFrame.interpolate.html

**Web.** Available at https://easyvista2015-my.sharepoint.com/personal/aait-bachir_
easyvista_com/_layouts/15/onedrive.aspx?id=%2Fpersonal%2Faait%2Dbachir%5Feasyvista%5Fcom%
2FDocuments%2FLab%2FPublicData&ga=1.

**Antivirus.** Available at https://easyvista2015-my.sharepoint.com/personal/aait-bachir_
easyvista_com/_layouts/15/onedrive.aspx?id=%2Fpersonal%2Faait%2Dbachir%5Feasyvista%5Fcom%
2FDocuments%2FLab%2FPublicData&ga=1.

## E    Additional experiments

In Table 6 we provide F1 score on the adjacencies. In contrast to Table 1 in the main text, this Table shows the performance of the algorithms on skeleton recovery which allows to illustrate the robustness of CBNB class to Assumption 4.

Table 6: Results obtained on the simulated data of Section 5.2 for the different structures with 1000 observations with non-Gaussian noise (top panel) and with Gaussian noise (bottom panel). We report the mean and the standard deviation of the F1 score on adjacencies. The best results are in blue bold and the second best results are in green bold.

|  | Diamond | Cyclic Diamond | Unf. Diamond | Adj. Unf. Diamond |
|---|---|---|---|---|
|  | Non-Gaussian noise | | | |
| NBCB-w | **0.95** $\pm$ 0.01 | 0.95 $\pm$ 0.01 | 0.94 $\pm$ 0.01 | **0.88** $\pm$ 0.01 |
| CBNB-w | **0.95** $\pm$ 0.01 | **0.96** $\pm$ 0.01 | 0.95 $\pm$ 0.01 | **0.88** $\pm$ 0.01 |
| NBCB-e | 0.86 $\pm$ 0.01 | 0.85 $\pm$ 0.01 | **0.97** $\pm$ 0.01 | **0.87** $\pm$ 0.01 |
| CBNB-e | 0.85 $\pm$ 0.01 | 0.86 $\pm$ 0.01 | **0.97** $\pm$ 0.01 | **0.87** $\pm$ 0.01 |
| GCMVL | 0.87 $\pm$ 0.01 | 0.92 $\pm$ 0.01 | 0.07 $\pm$ 0.01 | 0.06 $\pm$ 0.02 |
| PCMCI$^+$ | **0.95** $\pm$ 0.01 | **0.96** $\pm$ 0.01 | 0.95 $\pm$ 0.04 | **0.88** $\pm$ 0.01 |
| PCGCE | 0.85 $\pm$ 0.01 | 0.86 $\pm$ 0.01 | **0.97** $\pm$ 0.01 | **0.87** $\pm$ 0.01 |
| Dynotears | 0.09 $\pm$ 0.03 | 0.0 $\pm$ 0.0 | 0.0 $\pm$ 0.0 | 0.0 $\pm$ 0.0 |
| VarLiNGAM | **0.99** $\pm$ 0.01 | **0.97** $\pm$ 0.01 | **0.98** $\pm$ 0.01 | **0.87** $\pm$ 0.01 |
|  | Gaussian noise | | | |
| NBCB-w | **0.93** $\pm$ 0.01 | **0.93** $\pm$ 0.01 | 0.90 $\pm$ 0.01 | **0.87** $\pm$ 0.01 |
| CBNB-w | **0.96** $\pm$ 0.01 | **0.96** $\pm$ 0.01 | **0.94** $\pm$ 0.01 | **0.87** $\pm$ 0.01 |
| NBCB-e | 0.83 $\pm$ 0.01 | 0.83 $\pm$ 0.01 | 0.93 $\pm$ 0.01 | **0.85** $\pm$ 0.01 |
| CBNB-e | 0.85 $\pm$ 0.01 | 0.84 $\pm$ 0.01 | **0.97** $\pm$ 0.01 | **0.87** $\pm$ 0.01 |
| GCMVL | 0.88 $\pm$ 0.01 | 0.91 $\pm$ 0.01 | 0.02 $\pm$ 0.01 | 0.03 $\pm$ 0.01 |
| PCMCI$^+$ | **0.96** $\pm$ 0.01 | **0.96** $\pm$ 0.01 | **0.94** $\pm$ 0.01 | **0.87** $\pm$ 0.01 |
| PCGCE | 0.85 $\pm$ 0.01 | 0.84 $\pm$ 0.01 | **0.97** $\pm$ 0.01 | **0.87** $\pm$ 0.01 |
| Dynotears | 0.13 $\pm$ 0.05 | 0.0 $\pm$ 0.0 | 0.0 $\pm$ 0.0 | 0.0 $\pm$ 0.0 |
| VarLiNGAM | 0.91 $\pm$ 0.01 | 0.92 $\pm$ 0.01 | 0.93 $\pm$ 0.01 | 0.84 $\pm$ 0.01 |