# OpenReview forum: "Causal Discovery from Time Series with Hybrids of Constraint-Based and Noise-Based Algorithms"
_TMLR — Accepted by TMLR_

### Review · Reviewer_AuBy · 2024-02-23

**Summary Of Contributions:**

This paper presents a flexible framework for hybrid (noise- and constraint-based) causal discovery algorithms for time series, successfully combining existing methods to achieve improved performance.

**Audience:**

Yes

**Claims And Evidence:**

No

**Requested Changes:**

*Critical:* (in order of appearance)
- 3rd sentence, Section 2: be more specific here and later about "graph"---sometimes it's important that you're talking about a simple DAG, other times it's important to have bidirected edges and self loops, etc. Clarify when you're talking about which classes of graphs.
- It seems like Assumptions 1 and 2 are implicitly about DAGs? This should be mentioned explicitly here too instead of as a seemlingly independent assumption 5.
- Doesn't Assumption 1 for DAGs imply Causal Sufficency?
- The statement of Assumption 5 only mentions (uni)directed cycles, but for example Fig 2c is described as having a cycle even though it's a bidirected cycle.
- $\mathbb{N}^*$ in Assumption 6 is undefined?
- Definition 2 implies you can have multigraphs with directed 2-cycles, but the example summary graphs seem to use bidirected edges instead?
- Second sentence after Definition 2 about inferring summary graphs should be clarified---can the summary graph be directly deduced from the window causal graph, or not? Are additional assumptions only sometimes required?
- 4th line of page 7: can you justify why you expect this? otherwise better to avoid speculation.
- Doesn't possiblity (3) in Proposition 1 violate Assumption 1?
- In Definition 6, what does it mean to take the intersection of two paths?
- Why not use the cycle $Z_t$ -- $X_t$ -- $Y_t$ -- $W_t$ -- $Z_t$ instead of the the two 3-cycles over those vertices?
- Discussion after Table 4:
  - "n such cases, the true graph may not be fully retrieved, but it is guaranteed that if the algorithms infer
X → Y , then in the true graph, we can be certain that Y → X"; But this is trivial when assuming the correct causal order as in Proposition 1.
  - "the true graph may not be completely retrieved, but the inferred graph’s skeleton
will be accurate"; This sentence indicates $\iff$ but the corresponding entry in Table 4 uses $\implies$.

*Suggested:* (in order of appearance)
- paragraph spacing seems to disappear after Assumption 3 and for the rest of the paper?
- just before Section 3.2: "conditional independence" instead of "conditionally independent"?
- Show experimental results using nonparamteric CI tests; should only require substituting 1 or 2 lines of code and should improve performance. Maybe using nonlinear regression is similarly easy? Doing one or both of these on the real data would already be quite compelling, even if you don't generate nonlinear simulated data to apply it to.
- After Definition 6, you use "cycle group" and "cyclic group". Is there supposed to be a distinction?
- Second paragraph in Section 5.2: are these supposed to be superscripts or subscripts?
- Discuss runtime of CBNB and NBCB compared to just PCMCI+ or VLiNGAM

**Strengths And Weaknesses:**

*Strengths:*
- generally clearly written paper, with helpful explanation and examples
- explicit statement and discussion of assumptions
- sufficient experimental results that show the algorithms perform well

*Weaknesses:*
- assumptions are a bit confusing
- some imprecision/ambiguity in the technical details
- lacking discussion on runtime

---

> ### Author Response · Authors · 2024-03-07
> **Response: part 1**
>
> We wish to thank the Reviewer for his/her effort in providing very useful comments that helped us to improve the manuscript.
> Detailed responses to all the points raised by the Reviewer are reported below.
> We have corrected typos. $\mathbb{N}^*$ was referring to nonzero natural numbers, we refer now to $\mathbb{N}\setminus \{0\}$. The intersection of two paths was referring to the common edges between the two paths, we will clarify this.
> We will provide an updated pdf when considering all the reviews.
>
> First, we will answer the following questions:
>
> 1. 3rd sentence, Section 2: be more specific here and later about "graph"—sometimes it’s important that you’re talking about a simple DAG, other times it’s important to have bidirected edges and self loops, etc. Clarify when you’re talking about which classes of graphs
>
> 2. It seems like Assumptions 1 and 2 are implicitly about DAGs? This should be mentioned explicitly
> here too instead of as a seemingly independent assumption 5
>
> 3. Doesn’t Assumption 1 for DAGs imply Causal Sufficency?
>
> 4. The statement of Assumption 5 only mentions (uni)directed cycles, but for example Fig 2c is described
> as having a cycle even though it’s a bidirected cycle.
>
> Response to questions 1, 2, 3, 4 :
>
> A1. We agree that we should be more clear using graph notion. We followed this recommendation and clarified when we referred to DAGs and when we discussed other types of graphs.
>
> A2. We agree that in the present form Assumption 5 is confusing and we remove it. Instead, we add the acyclicity definition in the description of the DAG.
>
> A3.  We introduced the Causal Markov condition (Assumption 1) in a general form, assuming that a set of vertices $\mathbb{V}$ in a DAG $\mathcal{G}$ can represent observed and unobserved variables. For example, for observed variables $X$, $Y$ , let $Z$ be such that $X  \leftarrow Z \leftarrow Y$, but $Z$ can be observed or unobserved. We then introduced Causal sufficiency assumption over observed variables, which explicitly states that $Z$ should be observed. To avoid ambiguity, we will start with Causal Sufficiency and then introduce Causal Markov condition.
>
> A4. The notion of (uni)directed cycles, is only defined for DAGs (full time causal graph, window causal graphs and extended causal graphs) but not for summary causal graphs since in this paper we do not infer directly summary causal graph but rather we infer a window causal graph (or an extended summary causal graph) and then deduce the summary causal graph from it. As mentioned above,  we remove Assumption 5 and discuss acyclicity together with the introduction of the DAG. We explicitly discuss the possibility of self-loops or two edges in opposite direction (cycles of size 2), when we introduce summary causal graphs.

---

> ### Author Response · Authors · 2024-03-07
> **Response: part 2**
>
> In the following, we give a response to the following two questions:
>
> 6. Definition 2 implies you can have multigraphs with directed 2-cycles, but the example summary graphs seem to use bidirected edges instead?
>
> 7. Second sentence after Definition 2 about inferring summary graphs should be clarified---can the summary graph be directly deduced from the window causal graph, or not? Are additional assumptions only sometimes required?
>
> Response to questions 6, 7:
>
> A6. We thank the reviewer for pointing out the discrepancy between Definition 2 and the graphical representation of the summary causal graph.  We replace bidirected edges with two edges in the opposite direction for every figure related to the summary causal graph.
>
>
> A7. We agree that the second sentence after Definition 2  is not clear.  Yes, a summary causal graph can be directly deduced from a window causal graph (or an extended summary causal graph) without additional assumption. However, it is not possible to infer it directly from data without additional assumptions. For example,  NBCB$^{acyclic}$\citep{Assaad_2021} can discover the summary causal graph under the additional assumption that the summary causal graph is acyclic. We will clarify this in the text. In this paper, we focus on the first case.

---

> ### Author Response · Authors · 2024-03-07
> **Response:  part 3**
>
> In the following, we answer the following questions:
>
> 8. 4th line of page 7: can you justify why you expect this? otherwise better to avoid speculation.
>
> 9. Doesn't possibility (3) in Proposition 1 violate Assumption 1?
>
> 10. Why not use the cycle $Z_t --X_t--Y_t--W_t--Z_t$ instead of the the two 3-cycles over those vertices?
>
> 11. Discussion after Table 4:
>
>      (a) "in such cases, the true graph may not be fully retrieved, but it is guaranteed that if the algorithms infer $X \leftarrow Y$ , then in the true graph, we can be certain that $Y \leftarrow X$"; But this is trivial when assuming the correct causal order as in Proposition 1
>
>     (b) "the true graph may not be completely retrieved, but the inferred graph’s skeleton will be accurate"; This sentence indicates $\equiv$  but the corresponding entry in Table 4 uses $\implies$.
>
>
>
> Response to questions 8, 9, 10 and 11:
>
> A8. We thank the reviewer for this question and apologize for an unsupported claim. We remove this sentence as for the moment we can not prove fully this generalization. In consequence, we restrict Proposition 1 to the class of additive noise-based models. We also mention this restriction in Table 4 and corresponding discussion.
>
>
> A9. Possibility (3) in Proposition 1 points a difference between the inferred graph and the true graph but do not say anything about the true graph. So, Possibility (3) does not violate Assumption 1 about the true graph.  This difference happens when an (iterative) algorithm makes several mistakes that correspond to  possibility (2) and then because of these mistakes the Algorithm will make a mistake which will lead to possibility (3). This mistakes are related to violation of Assumption 3 (Adjacency faithfulness). In the example provided in the paper,  true WCG (Figure 4 (a)) satisfies Assumption 1 (Causal Markov condition), but due to adjacency faithfulness violation, $Z_{t-1}$ is removed from the set of parents of $X_t$ in the inferred graph, which leads to an erroneous edge $X_t -Y_t$ .
>
> A10.  We discuss two 3-cycles between $X_t, Y_t, Z_t$ and $Z_t, Y_t, W_t$ in the example of Figure 5(b), which have a common edge $Z_t - W_t$ as a motivation for introducing the undirected cycle groups. Later we introduce undirected cycle groups, which gives us the undirected cycle $Z_t- X_t - Y_t -W_t - Z_t$ which you proposed to use.
>
>
> A11(a) Table 4 summarizes the main results in the paper and provides a guide on how to interpret graphs inferred by different algorithms and under different assumptions.  In case of violation of Assumption 3, the phrase "in such cases, the true graph may not be fully retrieved, but it is guaranteed that if the algorithms infer $X \rightarrow Y$ , then in the true graph, we can be certain that $Y \leftarrow X$ " is a summary of Proposition 1 (there was a typo on the direction of the edge, that has been corrected). We will mention it clearly in the text.
>
> A11(b) We agree that the mentioned phrase is unclear and rephrased in the following way: "In this scenario,  algorithms are capable of inferring accurate skeleton of the graph, but not the orientations." It is also true, that we have an equivalence, but this is not the message that we are trying to put forward in this table.

---

> ### Author Response · Authors · 2024-03-07
> **Response: part 4**
>
> Here we answer the following questions:
>
> 12. Show experimental results using nonparametric CI tests; should only require substituting 1 or 2 lines of code and should improve performance. Maybe using nonlinear regression is similarly easy? Doing one or both of these on the real data would already be quite compelling, even if you don't generate nonlinear simulated data to apply it to.
>
> 13. Discuss runtime of CBNB and NBCB compared to just PCMCI+ or VLiNGAM.
>
>
> A12. We thank the reviewer for this idea. We are now running our method, as well as PCMCI and PCGCE, with nonparametric CI tests on the real datasets, and results will be provided in the updated pdf.
>
>
> A13. About the running time of CBNB and NBCB, we will add an analysis of the theoretical complexity in the worst case in each section, and also a numerical illustration.
> When considering $d$ time series, finding the causal order is of complexity $d^2 f(n,d)$ where $f(n,d)$ is the complexity of the user-specific regression method with time series length $n$.
> Then, pruning the graph by  conditional independence tests is of complexity $\frac{(d\tau)^2 (d\tau-1)^{k-1}}{(k-1)!}$,where $k$ represents the maximal degree of any vertex and each operation consists in conducting significance test to a conditional independence measure.
> Both CBNB and NBCB have the same complexity in the worst case, $d^2 f(n,d) + \frac{(d\tau)^2 (d\tau-1)^{k-1}}{(k-1)!}$ which corresponds for CBNB to one cluster.
> This is similar to  PCMCI+ and VLiNGAM, and experimental results illustrate this as well. We will provide an illustration in the experimental section.

---

### Review · Reviewer_NiR4 · 2024-03-07

**Summary Of Contributions:**

The paper presents two new hybrid methods for causal discovery from time series with instantaneous effects. Both methods combine noise-based (NB) and constraint-based approache. They differ mainly in the order in which the two are applied:
- NBCB first uses NB to learn a causal order of instantaneous effects, resulting in a fully connected and fully oriented graph; CB is then used to prune edges.
- CBNB first uses CB to learn a partially directed graph in which only lagged edges are directed and instantaneous ones undirected; NB is then used to orient the instantaneous edges.

Both methods rely on combining standard approaches for CB and NB in the respective order, subject to minor refinements to reduce unnecessary computations:
- In NBCB, the CB step is refined by only searching for conditioning sets among parents, rather than all adjacent nodes.
- In CBNB, the NB step is refined by applying NB not to the set of all nodes, but separately to a partition thereof into appropriately chosen subsets (so-called "undirected cycle groups").

It is shown that both methods return the correct graph if all required assumptions are satisfied (Thms. 1 and 2). Further, some results are presented for partially misspecified settings:
- For NBCB, if adjacency faithfulness is violated, then it can still be guaranteed that if $X\to Y$ in the inferred graph, then $Y\not\to X$ in the true graph (Prop. 1).
- For CBNB, if the SCM is not identifiable, adjacency in the inferred graph still implies adjacency in the true graph (Prop. 2).

Empirically, linear versions of the proposed hybrid methods are compared against Granger-, constraint-, score-, and noise-based baselines on synthetic, semi-synthetic, and real datasets. The results show that CBNB and NBCB are trade-offs between CB and NB, which may exhibit increases robustness to violations of some assumptions.

**Audience:**

Yes

**Broader Impact Concerns:**

Not applicable.

**Claims And Evidence:**

Yes

**Requested Changes:**

Please address the points listed under main weaknesses above.

I would also appreciate adressing/responding to the more minor points (most of which are easily fixed), though these are less critical in nature for the overall evaluation.

**Strengths And Weaknesses:**

### Strengths
- The paper is well-structured and easy to follow. The writing is good overall, except in some sections (see below). The extensive background section helps make the work accessible even to researchers without much prior exposure to causal discovery in temporal settings.
- The presented hybrid framework is conceptually simple and clean. The paper does a good job at keeping things simple, and not overcomplicating things. The figures are nicely made, and together with the running example, are very helpful to communicate the different steps of the method, and the newly introduced concepts.
- Some theoretical results show the soundness of the proposed methods. The misspecification results, while limited in scope, are an interesting first step (that could be expanded).
- The experimental section is rather extensive, considering both simulated and real data, and several well-established baselines.
- The paper does a good job at adopting a sober scientific tone, and not overclaiming on the results, clearly emphasising that there are trade-offs and highlighting both benefits and downsides of the proposed methods.

### Main Weaknesses
- The paper does not succeed at clearly and convincingly presenting a strong argument for when and why the proposed hybrid framework is beneficial compared to pure CB or NB methods. The misspecification results are a step in this direction, but they are quite limited in scope and appear relatively straight-forward: Prop. 1 seems to follow more-or-less directly from the assumption of a given correct causal order, and similarly Prop. 2 from the assumption of a conditional independence oracle. Wouldn't the same results also hold for purely NB or CB methods?
- In Sec. 4.2, there is an inconsistency in the definition and use of undirected cycle paths/groups: according to Defn. 5, an undirected cycle path must have length at least 3, and according to Defn. 6 an undirected cycle group consists of one or more undirected cycle paths. However, below Defn. 6, $(U_t, W_t)$ is referred to as an undirected cycle group, despite being of size 2. This inconsistency needs to be resolved. In particular, it needs to be guaranteed that each edge is a member of at least one undirected cycle group, so that all edges can be oriented by Algorithm 2. With the current Defns., it is not clear that this should be the case.
- The literature review in Sec. 3.1 and the description of results in Sec. 5 are somewhat disorganized and not very well-written. The former should be better structured and connected to the material in Section 2, and the latter should focus more on high-level takeaways and which results are expected or surprising. Generally, the paper would benefit from grammar checking/proofreading by a native speaker as throughout, indefinite and definite articles are often missing, which hurts the flow/readability.
- Only linear methods are considered in the experiments. While this is justified for the synthetic data which has linear ground-truth, linearity is violated for the semi-synthetic data, and most-likely for many of the real datasets. As a result, it is unclear to what extent the results on these data sets reflect differences between the methods, rather than misspecification effects. To address this, it would be good to also try a nonlinear (e.g., kernel) regression approach, or to run an ablation for linear methods applied to nonlinearly generated synthetic data (or both).

### Other comments, suggestions, and more minor points
- after Assumption 3, it would be nice to give an example or some intuition for why adjacency faithfulness is weaker than faithfulness; an example is given later in the paper that could potentially already be mentioned here.
-  After Defn. 3: "Remark that" --> "Note that" / "We remark that"
- I strongly suggest using paragraphs (empty line in latex) rather than lineskips to make the manuscript more readable.
- In the example after Defn. 4, it may be worth noting that the order among $(U_t,X_t)$ is not determined: i.e., one could also have an order in which $\pi(U_t)=4$ and $\pi(X_t)=5$. The current presentation may be slightly misleading in this regard, particularly the last phrase.
- According to  Defn. 1, the graph in Fig. 4(a) is not a WCG since the lag is only 1, is it?
- I like the clearly stated and references assumptions. However, when referring to Assumptions at various parts throughout the paper, it would be helpful to also give the name of the assumption: e.g., "when the *Identifiability* Assumption 4 is violated ..." as opposed to "when the Assumption 4 is violated ..." --- doing this would save the reader a lot of flicking back and forth through the manuscript to look up which assumption is meant.
-  The paper often refers to "WCG or ECG", but it is not entirely clear to what extent the two can be considered equivalent for all intents and purposes in the studied setting; could you perhaps add some elaboration on this to the manuscript?
- at the bottom of page 12, penultimate line, there appears to be a typo: "adjacent to $Y$" --> "adjacent to $W$"
- when introducing the -w and -e versions of NBCB and CBNB, it is not clear what "w" and "e" stand for; later it becomes clear that this indicates whether a WCG or ECG is learnt, but it would be helpful to clarify this directly when introducing the variants
- What is going on with Dynotears in Table 1? How can the catastrophic performance be explained? Might this be an implementation error? (the numbers really stand out from all others)
- in 5.3, please specify the distribution from which $o_y$ is "randomly assigned between 0 and 1", is it uniform? also, please explain what trophic levels are (or avoid the term altogether)
- In the third line of p.18 there appears to be a typo: "we can be certain that $Y\to X$" should probably read "we can be certain that $Y\not\to X$"


### Summary:
*Overall, a solid submission with somewhat limited contributions or novelty, but sound claims and no major flaws that cannot easily be fixed. Subject to minor revisions (see below), I see no objections to acceptance. However, I think that expanding on the theoretical comparison and benefits compared to non-hybrid methods, would help substantially strengthen the work.*

---

> ### Author Response · Authors · 2024-03-20
> **Response: part 1**
>
> We wish to thank the Reviewer for the useful comments and valuable feedback. Detailed responses to all the points raised by the Reviewer are reported below.  We have corrected the typos after Definition 3 and on pages 12 and 18. We have taken into account the Reviewer's suggestions by using paragraphs throughout the paper and specifying the name when we refer to the assumptions.  We added the description of the  -w and -e versions of NBCB and CBNB in the text.
>
> First, we will answer the following two questions:
>
> 1.  The paper does not succeed at clearly and convincingly presenting a strong argument for when and why the proposed hybrid framework is beneficial compared to pure CB or NB methods. The misspecification results are a step in this direction, but they are quite limited in scope and appear relatively straight-forward: Prop. 1 seems to follow more-or-less directly from the assumption of a given correct causal order, and similarly Prop. 2 from the assumption of a conditional independence oracle. Wouldn't the same results also hold for purely NB or CB methods?
>
> 2.  In Sec. 4.2, there is an inconsistency in the definition and use of undirected cycle paths/groups: according to Defn. 5, an undirected cycle path must have length at least 3, and according to Defn. 6 an undirected cycle group consists of one or more undirected cycle paths. However, below Defn. 6, (Ut,Wt) is referred to as an undirected cycle group, despite being of size 2. This inconsistency needs to be resolved. In particular, it needs to be guaranteed that each edge is a member of at least one undirected cycle group, so that all edges can be oriented by Algorithm 2. With the current Defns., it is not clear that this should be the case.
>
>
> Response to questions 1 and 2 :
>
> A1. The advantage of the proposed hybrid frameworks is that they have a trade-off performance between the constraint-based and noise-based algorithms.  In comparison to the constrained-based family, the proposed methods do not require faithfulness (but require weaker assumption adjacency faithfulness) and can recover the true causal graph.  The advantage of the proposed hybrid frameworks in comparison to noise-based methods is the more robust pruning of edges when the sample size is small [2]. These properties result in a trade-off performance for hybrid methods, balancing between constraint-based and noise-based algorithms. In addition, we considered in Proposition 1 and Proposition 2, the cases when some of the assumptions needed for the hybrid algorithms are violated to illustrate to practitioners what can be expected in each of the situations and help them (through Table 4) to choose a suitable class of methods. In this case, the hybrids are as robust as one of the corresponding noise-based or constraint-based parts. In the new version of the paper, we clarified this in the text.
>
> A2. We thank the reviewer for this remark. We agree that our definition is not clear as we used the term "path". We revised the definition of an undirected cycle path to ensure that each edge belongs to exactly one undirected cycle group. In the revised version of the paper, we changed the name of the undirected cycle path to the undirected cycle walk. Additionally, we defined the size of an undirected cycle walk as the number of consecutive edges on the walk, which is equal at least to two. Furthermore, we modified the examples in the text accordingly.
>
> [2] Daniel Malinsky and David Danks. Causal discovery algorithms: A practical guide. Philosophy Compass,
> 13(1) : e12470, 2018

---

> ### Author Response · Authors · 2024-03-20
> **Response: part 2**
>
> Here,  we answer the following questions:
>
> 3. The literature review in Sec. 3.1 and the description of results in Sec. 5 are somewhat disorganized and not very well-written. The former should be better structured and connected to the material in Section 2, and the latter should focus more on high-level takeaways and which results are expected or surprising. Generally, the paper would benefit from grammar checking/proofreading by a native speaker as throughout, indefinite and definite articles are often missing, which hurts the flow/readability.
>
> 4. Only linear methods are considered in the experiments. While this is justified for the synthetic data which has linear ground-truth, linearity is violated for the semi-synthetic data, and most-likely for many of the real datasets. As a result, it is unclear to what extent the results on these data sets reflect differences between the methods, rather than misspecification effects. To address this, it would be good to also try a nonlinear (e.g., kernel) regression approach, or to run an ablation for linear methods applied to nonlinearly generated synthetic data (or both).
>
> Response to the questions 3 and 4:
>
> A3. We have rewritten some parts in Sec 3.1 to refer to the assumptions and graphs mentioned in Sec 2, and we have highlighted the structure of the section. We have rewritten Sec 5, to highlight expected and surprising results. We did a thorough grammar check of the paper.
>
> A4.   We thank the reviewer for this comment. Running nonlinear approaches would be time-consuming for the semi-synthetic data, as there are 100 datasets for the two cases. Our aim was to understand the robustness of our method, with linear parameters, using some nonlinear model, where the nonlinearity can be controlled. Then, we study how the linearity is a good proxy for this modeling. However, we have noticed that we used $500$ time steps instead of available $1000$. We have corrected this error, which slightly improved results in Table 2. Nevertheless, we are now running nonlinear regression model (based on GP) and nonlinear test (based on kernels) for the real data (but excluding the Web and Antivirus datasets due to computational constraints). Results will be provided in the updated pdf, as well as the analysis.

---

> ### Author Response · Authors · 2024-03-20
> **Response: part 3**
>
> In the following, we answer the following questions:
>
> 5.  after Assumption 3, it would be nice to give an example or some intuition for why adjacency faithfulness is weaker than faithfulness; an example is given later in the paper that could potentially already be mentioned here
>
> 6.  In the example after Defn. 4, it may be worth noting that the order among  $(Ut,Xt)$ is not determined: i.e., one could also have an order in which  $\pi(Ut)=4$ and $\pi(Xt)=5$. The current presentation may be slightly misleading in this regard, particularly the last phrase.
>
> 7. According to Defn. 1, the graph in Fig. 4(a) is not a WCG since the lag is only 1, is it?
>
> 8.   The paper often refers to "WCG or ECG", but it is not entirely clear to what extent the two can be considered equivalent for all intents and purposes in the studied setting; could you perhaps add some elaboration on this to the manuscript?
>
> 9. What is going on with Dynotears in Table 1? How can the catastrophic performance be explained? Might this be an implementation error? (the numbers really stand out from all others)
>
> 10. in 5.3, please specify the distribution from which $o_y$ is "randomly assigned between 0 and 1", is it uniform? also, please explain what trophic levels are (or avoid the term altogether)
>
> Response to the questions 5,6,7,8,9, and 10:
>
> A5. We thank the reviewer for this suggestion. We added an example after Assumption 3, which illustrates the situation when the faithfulness assumption is violated, but the adjacency faithfulness is satisfied. In addition, we have added an example, when both assumptions are not satisfied.
>
> A6. We thank the reviewer for this remark. We have corrected the example after Definition 4.
>
> A7. We thank reviewer for this question. The window size of the WCG is equal to maximum temporal lag $\gamma +1$, which is inconsistent with the Figure 4(a). We have changed the Figure 4(a) and corresponding example.
>
> A8. We considered several causal graphs in this paper. The FTCG and corresponding representations under Consistency Assumption 6, a WCG and ECG are directed acyclic graphs, while an SCG can be cyclic. Moreover, an ECG and a WCG are  equivalent when the maximal temporal  lag $\gamma$ is equal to 1.  We consider causal discovery methods in this paper, that can infer either a WCG or an ECG. So, we provide theoretical results for both WCG and ECG and the proofs are equivalent as they are based on same the assumptions mentioned in the paper and on the acyclicty of the corresponding causal graph. We have included a more comprehensive description in the text. In addition, we have changed the order of definitions of the ECG and the SCG to enhance clarity.
>
> A9. We thank the reviewer for this question. We use the implementation of the Dynotears method provided in the library
> CausalNex \url{https://github.com/quantumblacklabs/causalnex}.  To verify the results of Dynotears in Table 1,  we re-run the experiments using the same package as before. We have obtained the same results a second time.  However, we think we can explain the poor performance of Dynotears on this simulated data.  As we mentioned in Section 3.1, Dynotears requires that marginal variance increase in accordance with the topological order of the WCG. We have verified that this condition is not satisfied in the simulated data, so it can lead to poor performance of Dynotears.
>
> A10. Yes, the $o_y$ is sampled uniformly between 0 and 1 (we sample from $U([0.05,0,95])$ in order to avoid including $0$ and $1$). We specified it in the text. We have also included an explanation of the trophic levels.

---

### Review · Reviewer_FAfh · 2024-03-09

**Summary Of Contributions:**

The authors propose a hybrid framework for the causal of time series that combines parts of noise-based and
constraint-based algorithms. Within this framework, they derive two classes of algorithms, NBCB and CBNB,
which they optimized to infer the causal graph from time series. The authors study theoretically to which extent
each class of algorithms is robust against assumption violation. The authors provide extensive simulation studies
and real data applications to illustrate the applicability of their approach and their method’s enhanced capabilities
against assumption violation compared to original methods.

**Audience:**

Yes

**Broader Impact Concerns:**

This paper does not cover sensitive topics.

**Claims And Evidence:**

Yes

**Requested Changes:**

-[Figure 4] Give more detailed reasons for Zt−1 ⊥⊥ Xt|{Xt−2, Zt−2} so that readers can better understand.

-[Undirected Cycle Group] How to understand the undirected cycle group based on Wt and Ut, which is
Wt − Ut in Figure 5. This path only contains two nodes, and this path does not meet the conditions in Definition
5 (Undirected Cycle Path): size(u) ≥ 3, so it is also The conditions in Definition 6 (Undirected Cycle Group)
obtained based on Definition 5 are not met, that is: according to my understanding, the undirected path Wt − Ut
does not belong to an undirected cycle group. In other words, is this a special undirected cycle group?

-[NB$1^{\prime}$] "Find the causal order between instantaneous groups of nodes
$\mathbb{I}_t \subseteq \mathbb{V}_t $
by recursively performing regression and independence tests between the predictors and residuals (noise)." " Given the output of the CB1 step, CBNB searches for all undirected cyclic groups. Then for each undirected cyclic group $\mathbb{C}$, CBNB uses NB$1^{\prime}$ to find the causal order $\pi$ between all the nodes belonging to undirected cyclic group $\mathbb{C}$." The first sentence describes using NB$1^{\prime}$ to find the causal order between instantaneous groups, and the second sentence describes using NB$1^{\prime}$ to find the causal order $\pi$ between all the nodes belonging to undirected cyclic group $\mathbb{C}$. This is very different, and the author needs to check these descriptions.

-The network in the experiment of Table 1 has too few instantaneous nodes.

-It is necessary to explain the complexity of the algorithm.

-  Discussion: "In such cases, the true graph may not be fully retrieved, but it is guaranteed that if the algorithms infer
$X \rightarrow Y$, then in the true graph, we can be certain that \underline{$Y \rightarrow X$}."  This should be $Y \not\rightarrow X$, not \underline{$Y \rightarrow X$}.

**Strengths And Weaknesses:**

---Strengths

The paper is well written. It contains a good literature review. I appreciate the authors’effort to discuss all
the necessary theoretical points that made it easy to understand their approach. Also, the plots are quite useful to
understand the results.

When Assumption 3 (Adjacency Faithfulness) is violated, the NBCB class of algorithms can still obtain valid
causal information.

When Assumption 4 (Identifiable Functional Model) is violated, the CBNB class of algorithms
can guarantee that the learned skeleton of the WCG or the ECG is accurate.

Experiments show that this algorithm has a certain degree of robustness.

---Weaknesses

When Assumptions 3 (Adjacency Faithfulness) and 4 (Identifiable Functional Model) are violated at the
same time, the method proposed in this paper is not reliable. The algorithm’s robustness in this paper is limited,
and it would be meaningful if it could be expanded to be stronger.

If the authors give an example of a situation where the faithfulness assumption is violated but the adjacency
faithfulness is satisfied, then explain the situation where adjacency faithfulness is violated. It will help readers
better understand the content of the paper.

In CBNB, the adjacency faithfulness assumption was used when learning the skeleton, and the V-structure
was learned. Why can’t it be used for orientation?

- When the noise is Gaussian, the experimental results in Table 1 are not as good as in the non-Gaussian case.
Can the author explain the reason for this phenomenon?

Why in the case of Gaussian noise in Table 1, in Diamond’s learning results, NBCB and CBNB are far behind
PCMCI+ and GCMVL? What is the specific reason?

What is the reason for the poor experimental results in Table 2?

If the author can add experiments comparing learning ECG and WCG, it will increase the completeness of
the theory in the article.

---

> ### Author Response · Authors · 2024-03-20
> **Response: part 1**
>
> We wish to thank the Reviewer for constructive and insightful comments. Detailed responses to all the points raised by the Reviewer are reported below. We have corrected the typo in the Discussion and provided more details for the example in Figure 4.
>
> First, we answer the following questions:
> 1. When Assumptions 3 (Adjacency Faithfulness) and 4 (Identifiable Functional Model) are violated
> at the same time, the method proposed in this paper is not reliable. The algorithm’s robustness in
> this paper is limited, and it would be meaningful if it could be expanded to be stronger.
> 2. If the authors give an example of a situation where the faithfulness assumption is violated but
> the adjacency faithfulness is satisfied, then explain the situation where adjacency faithfulness is
> violated. It will help readers better understand the content of the paper.
> 3. In CBNB, the adjacency faithfulness assumption was used when learning the skeleton, and the
> V-structure was learned. Why can’t it be used for orientation?
>
> Response to questions 1, 2, and 3:
>
> A1. We thank the reviewer for this comment.  In general,  without  Assumptions 3 and 4, the causal graph is not identifiable from observational data. In the literature, people have focused on different families, among them constraint-based methods (which rely on the faithfulness assumption) and noise-based methods (which rely on identifiable functional model). We have succeeded, in this paper, in lightening those assumptions by proposing a hybrid approach and providing two correct methods. We also give robustness results when violating some assumptions. We argue that it can not be expanded, in the sense that if we remove all the assumptions, the graph cannot be identifiable (using our methods or other methods).
>
> A2. We thank the reviewer for this suggestion.  We added an example when "the faithfulness assumption is violated, but the adjacency faithfulness is satisfied" in Section 2.1. We also provide an example when both assumptions are not satisfied.
>
> A3. We have mentioned in the description of the constraint-based algorithms, that steps  CB1 and CB2 need the faithfulness assumption, as the adjacency faithfulness assumption is not sufficient. Indeed, using only the adjacency faithfulness assumption it is not possible to reliably detect the V structures. We can consider the example provided by [1], if we have a $A \rightarrow B \rightarrow C $, such that $A ⫫  C$ and $A ⫫ C \mid B$, then the PC algorithm would remove an edge between $A$ and $C$ in the first step as $A ⫫ C$. Secondly, as $B$ was not used for pruning this edge, the PC algorithm will infer that $A \rightarrow B\leftarrow C$, which is a mistake.  In the paper, we add the reference to [1].
>
>
>
> [1] Joseph Ramsey, Peter Spirtes, and Jiji Zhang. Adjacency-faithfulness and conservative causal inference.
> In Proceedings of the Twenty-Second Conference on Uncertainty in Artificial Intelligence, UAI’06, pp.
> 401–408, Arlington, Virginia, USA, 2006. AUAI Press. ISBN 0974903922.

---

> ### Author Response · Authors · 2024-03-20
> **Response: part 2**
>
> In the following, we give a response to the following three questions:
>
> 4. When the noise is Gaussian, the experimental results in Table 1 are not as good as in the
> non-Gaussian case. Can the author explain the reason for this phenomenon? Why in the case of
> Gaussian noise in Table 1, in Diamond’s learning results, NBCB and CBNB are far behind PCMCI+
> and GCMVL? What is the specific reason?
>
> 5.  What is the reason for the poor experimental results in Table 2?
>
> 6.  If the author can add experiments comparing learning ECG and WCG, it will increase the completeness of the theory in the article.
>
> Response to questions 4, 5, and 6:
>
> A4. We are considering a hybrid of constraint-based and noise-based causal discovery methods, and in practice, our implementation relies on VarLiNGAM, which assumes that the noise is not Gaussian. This explains why the results in general, for NBCB, CBNB, and VarLiNGAM, are better in the non-Gaussian case. Note that this is not the case for PCMCI$^+$ and PCGCE, which does not relate on any distribution assumption.  This also explains why NBCB and CBNB are far behind PCMCI$^+$ and GCMVL. The orientation step, deduced from the causal order given by the noise step, is not reliable. However, the main message here is that our hybrid methods have a trade-off performance between constraint-based and noise-based methods even when the non-Gaussianity assumption is violated. We discuss this in Section 5.
>
> A5. We thank the reviewer for this question.  Semi-synthetic data are nonlinear by construction, as given in Equations preys/predators. We stick to linear modelling (through the linear test and the linear regressor) to understand the robustness of the method with respect to nonlinearity. However, we have noticed that we used $500$ time steps instead of the available $1000$. We have corrected this error, which slightly improved results in Table 2. In addition, in real data, we provide in the new version of the paper some results using (through a nonlinear test and a nonlinear regressor).
>
> A6. We thank the reviewer for this comment. We evaluate the results of the experiments on the SCG. We did not provide the evaluation on the ECG and the WCG as some methods we are comparing with, discover an ECG and others discover a WCG, so we compare the results using the SCG because it can be deduced from the output of any method.

---

> ### Author Response · Authors · 2024-03-20
> **Response: part 3**
>
> In the following, we give a response to the following questions:
>
> 7. [Undirected Cycle Group] How to understand the undirected cycle group based on Wt and Ut, which
> is Wt - Ut in Figure 5. This path only contains two nodes, and this path does not meet the conditions in
> Definition 5 (Undirected Cycle Path): size(u) ≥ 3, so it is also The conditions in Definition 6 (Undirected
> Cycle Group) obtained based on Definition 5 are not met, that is: according to my understanding, the
> undirected path Wt - Ut does not belong to an undirected cycle group. In other words, is this a special
> undirected cycle group?
>
> 8. [NB1'] Find the causal order between instantaneous groups of nodes $It\subseteq Vt$ by recursively performing regression and independence tests between the predictors and residuals (noise). Given the output of the CB1 step, CBNB searches for all undirected cyclic groups. Then for each undirected cyclic group C, CBNB uses NB1' to find the causal order $\pi$ between all the nodes belonging to undirected cyclic group C. The first sentence describes using NB1' to find the causal order between instantaneous groups, and the second sentence describes using NB1' to find the causal order  $\pi$ between all the nodes belonging to undirected cyclic group C. This is very different, and the author needs to check these descriptions.
>
> 9. The network in the experiment of Table 1 has too few instantaneous nodes.
>
> 10.  It is necessary to explain the complexity of the algorithm.
>
> Response to questions 7, 8, 9,and 10:
>
> A7. We thank the reviewer for this point. Our definition of the undirected cycle group, and more precisely the examples around it, were not so clear. We modified this part around Definition 5 and Definition 5 as well. We also change the name of the undirected cycle path to the undirected cycle walk. Additionally, we defined the size of an undirected cycle walk as the number of consecutive edges on the walk, which is equal at least to two.
>
> A8. We thank the reviewer for this question. The first mentioned sentence contains a typo,  NB1$^\prime$ is used to find the causal order between instantaneous nodes within the same undirected cycle group. NB1$^\prime$ is independently applied to each undirected cycle group.
>
> A9.  The number of instantaneous edges is random, as stated in the data-generating process at the beginning of Section 5.2. The number of nodes is kept to 4 because this section of simulated data is intended to illustrate how the methods work on building blocks, especially when testing the violation of assumptions. We argue that this extensive simulation study is interesting enough, even with only 4 nodes, and performance on larger networks is shown in Sections 5.3 and 5.4.
>
> A10. We have added an analysis of the theoretical complexity in the worst case in each section, and also a numerical illustration.
> When considering $d$ time series, finding the causal order is of complexity $d^2 f(n,d)$ where $f(n,d)$ is the complexity of the user-specific regression method with time series length $n$.  Then, pruning the graph by  conditional independence tests is of complexity $\frac{(d\tau)^2 (d\tau-1)^{k-1}}{(k-1)!}$, where $k$ represents the maximal degree of any vertex and each operation consists in conducting significance test to a conditional independence measure.  Both CBNB and NBCB have the same complexity in the worst case, $d^2 f(n,d) + \frac{(d\tau)^2 (d\tau-1)^{k-1}}{(k-1)!}$ which corresponds for CBNB to the case where there is only one undirected cycle group.  This is similar to  PCMCI+ and VLiNGAM, and experimental results illustrate this as well. We will provide an illustration in the experimental section.

---

### Author Response · Authors · 2024-03-22
**The new version of the paper.**

We are very thankful for your review.   We have responded to all your comments and have uploaded a revised version of the paper taking into acount your suggestions.  The main changes are highlighted in green. If anything is not clear or you have any further question, we would be happy to disccus them and make necessary changes.

---

> ### Comment · Reviewer_AuBy · 2024-03-25
>
> I'm not sure which of my requested changes were addressed/acknowledged. I see that some (but not others?) have been taken into account in the revised draft. Could the authors please clarify?

---

> > ### Author Response · Authors · 2024-03-25
> >
> > We provide an answer to each of your points just below your review (organized by themes through 4 parts).
> > Can you see it?
> > If there is anything we missed, we apologize and would be happy to answer.

---

> > > ### Comment · Reviewer_AuBy · 2024-03-25
> > >
> > > I see the 3-part responses to the other two reviews, but I see no response to mine. Could you check who the responses are visible to, or maybe try reposting it?

---

> > > > ### Author Response · Authors · 2024-03-25
> > > >
> > > > We apologize that it was not visible to you. We adjusted the visibility settings.  Could you kindly confirm if it is now visible to you?

---

> > > > > ### Comment · Reviewer_AuBy · 2024-03-25
> > > > >
> > > > > Yes, I see it now, thanks!

---

### Decision · Action_Editor_Zkc1 · 2024-04-14

**Recommendation:** Accept as is

**Comment:**

This paper presents a novel framework that merges constraint-based and noise-based methods to learn causal graphs from observational time series comprehensively. Based on which type of method is used firstly, two classes of methods are proposed in the paper. The paper furnishes theoretical assurances for each class, contingent upon the satisfaction of all assumptions, and delineates certain properties when assumptions are violated. To substantiate the effectiveness of the framework, two algorithms from each class are rigorously tested on simulated data, realistic ecological data, and datasets sourced from various real-world applications.

The reviewers raised several concerns on the the technical details and presentation, and these concerns were well addressed in the rebuttal. The updated manuscript was recommended for acceptance by all reviewers. After reviewing the paper and tracking the review process, I am in full agreement with the recommendation made by the reviewers.

**Audience:**

Yes

**Claims And Evidence:**

Yes